# Near-Optimal Regret Bounds for Multi-batch Reinforcement Learning

**Zihan Zhang***     **Yuhang Jiang**[†]     **Yuan Zhou**[‡]     **Xiangyang Ji** [§]

## Abstract

In this paper, we study the episodic reinforcement learning (RL) problem modeled by finite-horizon Markov Decision Processes (MDPs) with constraint on the number of batches. The multi-batch reinforcement learning framework, where the agent is required to provide a time schedule to update policy before everything, which is particularly suitable for the scenarios where the agent suffers extensively from changing the policy adaptively. Given a finite-horizon MDP with $S$ states, $A$ actions and planning horizon $H$, we design a computational efficient algorithm to achieve near-optimal regret of $\tilde{O}(\sqrt{SAH^3K\ln(1/\delta)})$[5] in $K$ episodes using $O\left(H + \log_2\log_2(K)\right)$ batches with confidence parameter $\delta$. To our best of knowledge, it is the first $\tilde{O}(\sqrt{SAH^3K})$ regret bound with $O(H + \log_2\log_2(K))$ batch complexity. Meanwhile, we show that to achieve $\tilde{O}(\text{poly}(S, A, H)\sqrt{K})$ regret, the number of batches is at least $\Omega\left(H/\log_A(K) + \log_2\log_2(K)\right)$, which matches our upper bound up to logarithmic terms.

Our technical contribution are two-fold: 1) a near-optimal design scheme to explore over the unlearned states; 2) an computational efficient algorithm to explore certain directions with an approximated transition model.

## 1 Introduction

In reinforcement learning (RL), the learning agent interacts with the environment to maximize the total reward by making sequential decisions. The agent typically has to achieve two seemingly very different goals: to try as many actions and reach as many states as possible so as to learn more information about the environment (a.k.a. *exploration*) and to follow the policy that collects the high rewards according to the learned information (a.k.a. *exploitation*). To address this exploration-exploitation dilemma and achieve the near-optimal regret bounds, the agent usually needs to adjust his/her strategies *adaptively* based on the historical trajectories and make frequent policy changes [Azar et al., 2017, Zanette and Brunskill, 2019, Zhang et al., 2020].

On the other hand, however, too much adaptivity requirement usually leads to lower level of parallelism, impeding the large-scale deployment of the RL algorithms (which is often in a distributed manner). Frequent policy updates also suffer the cost of re-deploying policies in many practical applications. For example, in medical domains, it often requires complete discussion among many experts to change the treatment plans, which is not affordable in terms of both time and monetary cost [Lei et al., 2012, Almirall et al., 2012, 2014]; in RL for hardware placement [Mirhoseini et al., 2017], rewriting the program into the hardware for too many times is strongly discouraged. Similar

---

*Department of Automation, Tsinghua University, `zihan-zh17@mails.tsinghua.edu.cn`

[†]Department of Automation, Tsinghua University, `jiangyh19@mails.tsinghua.edu.cn`

[‡]Yau Mathematical Sciences Center & Department of Mathematical Sciences, Tsinghua University, `yuan-zhou@tsinghua.edu.cn`

[§]Department of Automation, Tsinghua University, `xyji@tsinghua.edu.cn`

[5]$\tilde{O}(\cdot)$ hides logarithmic terms of $(S, A, H, K)$

36th Conference on Neural Information Processing Systems (NeurIPS 2022).

challenges also arise in applying RL to personalized recommendation system [Yu et al., 2019] and database optimization [Krishnan et al., 2018].

In such cases, the learning agent should minimize the number of policy switches while keeping the regret affordable. Bai et al. [2019] first proposed the provably efficient RL algorithms with low switching costs under the $Q$-learning algorithmic framework together with the lazy update techniques. However, their method needs to actively monitor the data in real time to determine whether a policy change is to be initiated. In other words, although the number of policy switches by [Bai et al., 2019] is low, the (usually long) time periods when the same policy is used still cannot be parallelized due to the policy-change trigger in their algorithms which is intrinsically sequential.

In order to address this problem, we propose and study under the framework of *multi-batch RL*, where the learning agent has to determine the number of batches and length of each batch before the learning process starts,[6] and uses as few batches as possible to achieve a low regret. Multi-batch RL algorithms can be easily deployed in a distributed fashion as the episodes during the same batch can be easily and fully parallelized. The idea of batch learning is also being widely practiced. For example, in medical trials, the medical center usually collects the data during a fixed time period among a batch of patients and then designs the experiment for the next phase based on the learned information in previous phases [Lei et al., 2012, Almirall et al., 2012, 2014].

Formally, we define multi-batch RL and *batch complexity* as below.

**Definition 1** (Multi-Batch RL with complexity $M$)**.** *The agent determines a group of lengths $\{t_m\}_{m=1}^{M}$ such that $\sum_{m=1}^{M} t_m = K$ before the learning process starts. For $m = 1, 2, \ldots, M$, the agent sets a policy $\pi^m$ and then follows $\pi^m$ for $t_m$ episodes.*

We highlight that an upper bound for batch complexity implies the same upper bound for global switching cost, since each policy switch means a new batch. It is also worth noting that the proposed batch RL framework is fully parallelizable during each batch for the applications where dataset comes in batch (e.g., clinical trial). Like other RL settings, we have the natural and interesting question:

**Question 1.** *Is it possible to achieve near optimal batch complexity, while keeping the regret $\tilde{O}(\sqrt{SAH^3K})$.*

We provide a positive answer for Question 1, which we state as below.

**Theorem 1.** *Let[7] $\iota = \ln(2/\delta)$. For any episodic MDP, with probability $1 - \delta$, under Algorithm 1 the regret in $T$ episodes is bounded by*

$$\text{Regret}(T) \leqslant \tilde{O}\left( \sqrt{SAH^3K\iota^2} + S^{\frac{15}{4}}A^{\frac{9}{8}}H^{\frac{17}{8}}\iota^{\frac{5}{8}}K^{\frac{3}{8}} + S^{\frac{19}{4}}A^{\frac{13}{4}}H^{\frac{33}{4}}\iota K^{\frac{1}{4}} + S^{\frac{11}{2}}A^{\frac{9}{2}}H^{\frac{17}{2}}\iota \right),$$

*and the batch complexity is bounded by $O(H + \log_2 \log_2(K))$. Moreover, the computational cost of Algorithm 1 is $\tilde{O}(S^4AHK^3 + S^3A^2H^2K^3)$.*

On the other hand, we show a lower bound of batch complexity as below.

**Theorem 2.** *For any algorithm with $O(\text{poly}(S, A, H)\sqrt{K})$ regret bound, the batch complexity is at least $\Omega(H/\log_A(K) + \log_2 \log_2(K))$.*

Compared to the lower bound of $\Omega(\log_2 \log_2(K))$ in [Gao et al., 2019] for multi-armed bandit problem, additional $\Omega(H/\log_A(K))$ batches are required to explore the structure of the MDP.

Due to space limitation, we defer the full proofs of Theorem 1 and Theorem 2 to Appendix D and Appendix B respectively.

**Our contribution.** We propose the framework of multi-batch RL, and first achieve $O(H + \log_2 \log_2(K))$ sample complexity bound with the near-optimal $\tilde{O}(\sqrt{SAH^3K\iota})$ regret bound with an efficient algorithm. We also prove that for any algorithm with $O(\text{poly}(S, A, H)\sqrt{K})$ regret, the global switching cost is at least $\Omega(H/\log_A(K) + \log_2 \log_2(K))$, which implies a nearly matching lower bound of $\Omega(H/\log_2(K) + \log_2 \log_2(K))$ for the batch complexity. We also note that the $O(H + \log_2 \log_2(K))$ batch complexity implies an $O(H + \log_2 \log_2(K))$ bound for the global switching cost, which is also a near optimal upper bound.

---

[6]In contrast, Bai et al. [2019] can update the policy at any time.

[7]Throughout the paper we use $\iota$ to denote $\ln(2/\delta)$.

## 2 Related Works

**Bandit algorithms with limited adaptivity.**  Bandit problem with low switching cost is widely studied in past decades [Cesa-Bianchi et al., 2013, Perchet et al., 2016, Gao et al., 2019, Simchi-Levi and Xu, 2019]. Cesa-Bianchi et al. [2013] showed an $\tilde{\Theta}(K^{\frac{2}{3}})$ regret bound under adaptive adversaries and bounded memories. Perchet et al. [2016] proved a regret bound of $\tilde{\Theta}(K^{\frac{1}{1-2^{1-M}}})$ for the two-armed bandit problem within $M$ batches, and later Gao et al. [2019] extended their result to the general $A$-armed case. Besides the setting of classical multi-armed bandit problem, other settings has also been studied, e.g., multinomial bandit problem [Dong et al., 2020] and linear bandit problem [Ruan et al., 2020].

**Episodic reinforcement learning with low switching cost.**  For model-based algorithms, by doubling updates, the global switching cost is $O(SAH\log_2(K))$ while keeping the regret $\tilde{O}(\sqrt{SAKH^3})$Azar et al. [2017]. For model-free algorithms, Bai et al. [2019] first studied RL with low switching cost. They proposed a $Q$-learning algorithm with lazy update to achieve $\tilde{O}(\sqrt{SAKH^4})$ regret bound and $O(SAH^3\log(K/A))$ local switching cost. Recently Zhang et al. [2020] established a better regret bound of $\tilde{O}(\sqrt{SAKH^3})$ and $O(SAH^2\log(K/A))$ local switching cost. Besides, Gao et al. [2021] generalized the problem to Linear RL, and established a regret bound of $\tilde{O}(\sqrt{d^3H^4K})$ with $O(dH\log(K))$ global switching cost. Recent work Qiao et al. [2022] achieved $O(HSA\log_2\log_2(K))$ switching cost and $\tilde{O}(\text{poly}(S,A,H)\sqrt{K})$ regret with a computational inefficient algorithm.

**Regret minimization for reinforcement learning.**  There is a long line of works devoting to regret minimization for RL problem [Kakade, 2003, Jaksch et al., 2010, Bartlett and Tewari, 2009, Dann et al., 2019, Azar et al., 2017, Jin et al., 2018, Zanette and Brunskill, 2019, Zhang and Ji, 2019, Zhang et al., 2020, Li et al., 2020, Zhang et al., 2021]. For tabular setting, near optimal regret bound of $\tilde{O}(\sqrt{SAH^3T})$ has been established by [Azar et al., 2017, Zanette and Brunskill, 2019, Zhang et al., 2020] for both model-based and model-free algorithms. However, fewer algorithms focused on the setting of multi-batch RL.

## 3 Preliminaries

**Episodic reinforcement learning.**  $M = \langle \mathcal{S}, \mathcal{A}, r, P, s_1 \rangle$, where $\mathcal{S} \times \mathcal{A}$ is the discrete state-action space, $r = \{r_h(s,a)\}_{(s,a)\in\mathcal{S}\times\mathcal{A},h\in[H]}$ is the known[8] reward function, $P = \{P_h(s,a)\}_{(s,a)\in\mathcal{S}\times\mathcal{A},h\in[H]}$ is the unknown transition model and $s_1$ is the fixed initial state[9]. We assume that the reward function $r_h(s,a) \in [0,1]$ for any $(h,s,a)$. In each episode, the agent starts at $s_1$, then takes actions and transits to the next state step by step, and finally conducts the trajectory $\{(s_h,a_h,s_{h+1})\}_{h=1}^H$. The target of the agent is to maximize the accumulative reward function $\sum_{h=1}^H r_h(s_h,a_h)$.

A policy $\pi$ can be viewed as a series of mappings $\{\pi_h\}_{h=1}^H$ where $\pi_h : \mathcal{S} \to \Delta^{\mathcal{A}}$ maps $s_h$ to a distribution over the action space at the $h$-th step, where $\pi_h(a|s)$ is the probability taking action $a$ at state $s$ of the $h$-th horizon.

Given a policy $\pi$, the (optimal) $Q$-function and value function are given by

$$Q_h^\pi(s,a) = \mathbb{E}_\pi\left[\sum_{h'=h}^H r_{h'}(s_{h'},a_{h'})\Big|(s_h,a_h)=(s,a)\right]; \qquad Q_h^*(s,a) = \sup_{\pi\in\Pi} Q_h^\pi(s,a);$$

$$V_h^\pi(s) = \mathbb{E}_\pi\left[\sum_{h'=h}^H r_{h'}(s_{h'},a_{h'})\Big|s_h=s\right]; \qquad V_h^*(s) = \max_a Q_h^*(s,a).$$

---

[8]This is a common assumption since the uncertainty of reward function is dominated by that of the transition model.

[9]The more general case, where the agent starts from a fixed initial distribution, could be reduced to our setting by increasing $H$ by 1

Let $\pi^{(k)}$ denote the policy in the $k$-th episode. Then the regret is given by

$$\text{Regret}(K) := \sum_{k=1}^{K} (V_1^*(s_1) - V_1^{\pi^{(k)}}(s_1)). \tag{1}$$

**Notations**  In this paper, we use $\mathbb{E}_{\pi,p}[\cdot]$ ($\mathbb{P}_{\pi,p}[\cdot]$) to denote the expectation (probability) following policy $\pi$ under transition model $p$. In particular, $\mathbb{E}_{\pi}[\cdot](\mathbb{P}_{\pi}[\cdot])$ denotes the expectation (probability) following $\pi$ under the true transition model $P$. We define the general value function

$$W^{\pi}(r',p) = \mathbb{E}_{\pi,p}\left[\sum_{h=1}^{H} r'_h(s_h, a_h)\right].$$

We use $\mathbf{1}$ to denote the $S$-dimensional vector $[1, 1, \ldots, 1]^{\top}$ and $\mathbf{1}_{h,s,a}$ to denote the reward function $r'$ such that $r'_{h'}(s', a') = \mathbb{I}[(h, s, a) = (h', s', a')]$. We also define $\{d_h^{\pi}(s,a)\}_{(s,a,h)}$ be the occupancy distribution of $\pi$. That is, $d_h^{\pi}(s,a) = \mathbb{E}_{\pi}[\mathbb{I}[(s_h, a_h) = (s,a)]]$. $\Delta^d$ is used to denote the $d$-dimensional simplex. For two vector $x, y$ with the same dimension, we write $x^{\top}y$ as $xy$ for convenience. For $p \in \Delta^S$ and $v \in \mathbb{R}^S$, we define $\mathbb{V}(p, v) = pv^2 - (pv)^2$. For $N \geqslant 1$, we use $[N]$ to denote the set $[1, 2, \ldots, N]$.

# 4  Technique Overview

In this section, we first introduce the policy elimination framework, which enjoys the near-optimal batch complexity. Then we summarize the technical challenges to achieve the near-optimal regret bound efficiently under this framework. At last, we introduce our major technical contributions.

## 4.1  Policy Elimination Framework

Following the methods in multi-batch bandit learning Perchet et al. [2016], Gao et al. [2019], we construct our main algorithm using policy elimination. Like most model-based reinforcement learning methods, we maintain a confidence region $\mathcal{P}$ for the transition model, where the true transition model $P \in \mathcal{P}$ with high probability. Before each batch starts, for a policy $\pi$ and a reward function $u$, by extended value iteration (See Algorithm 5 in Appendix C.2), we are able to compute the confidence interval $[L^{\pi}(u, \mathcal{P}), U^{\pi}(u, \mathcal{P})]$ for the value function of $\pi$, where

$$U^{\pi}(u, \mathcal{P}) := \max_{p' \in \mathcal{P}} W^{\pi}(u + \mathbf{1}_z, p'); \qquad L^{\pi}(u, \mathcal{P}) := \min_{p' \in \mathcal{P}} W^{\pi}(u, p'). \tag{2}$$

Here $z$ is a virtual state for the *infrequent* state-action-state triples (See Function `clip` in Algorithm 2). The reason why we give reward 1 for $z$ in computing the upper confidence bound is to encourage exploration to these *infrequent* state-action-state triples.

By policy elimination we get $\Pi(r, \mathcal{P}) = \left\{\pi \big| U^{\pi}(r, \mathcal{P}) \geqslant \sup_{\pi'} L^{\pi'}(r, \mathcal{P})\right\}$ as the set of survived policies. The next step is to choose a policy $\pi \in \Pi(r, \mathcal{P})$ and execute $\pi$ in the current batch. Defining $\mathcal{P}^m$ to be the confidence region for the transition model after the $m$-th batch and $\text{gap}^{m+1} = \max_{\pi \in \Pi(r, \mathcal{P}^m)}(U^{\pi}(r, \mathcal{P}^m) - L^{\pi}(r, \mathcal{P}^m))$, the regret in the $m + 1$-th batch could be bounded by $t^{m+1}\text{gap}^{m+1}$. Therefore, the main task is to design efficient exploration policy to reduce $\text{gap}^m$ for each $1 \leqslant m \leqslant M$.

## 4.2  Technical Challenges

Following the policy elimination framework above, we have two major challenges to achieve the near-optimal regret bound with an efficient algorithm.

**Difficulty in exploration**  Fix the reward function $r$ and confidence region $\mathcal{P}$. To construct tight confidence interval for every policy $\pi \in \Pi(r, \mathcal{P})$, we need to find a policy $\pi \in \Pi(r, \mathcal{P})$ to collect enough samples for each $(h, s, a)$. To address the problem, Qiao et al. [2022] proposed an algorithm named APEVE, which learns each $(h, s, a)$ triple independently. More precisely, for each $(h, s, a) \in [H] \times \mathcal{S} \times \mathcal{A}$, the algorithm searches for a policy $\pi^{h,s,a}$ to maximize the probability of visiting $(h, s, a)$

over $\Pi(r, \mathcal{P})$, and then execute $\pi^{h,s,a}$ to collect samples for $(h, s, a)$. However, this algorithm might be inefficient in sampling, since different horizon-state-action triples may match along with the same exploration policy. As shown in Qiao et al. [2022], the regret bound might be sub-optimal with this algorithm. Therefore, to achieve the near-optimal regret bound, we need to design a new exploration strategy to utilize the correlationship among different horizon-state-action triples.

**Difficulty in efficient implementation**  Because the policy set $\Pi(r, \mathcal{P})$ might have exponential size, naive enumeration is not applicable to searching for a good exploration policy. As a consequence, it requires additional efforts to study the structure of $\Pi(r, \mathcal{P})$. For example, when $r = 0$, $\Pi(r, \mathcal{P})$ is the set of all possible policies. In this case, we can use extended value iteration (See Algorithm 5) to find the policy which visits $(h, s, a)$ most frequently.

### 4.3  Key Techniques

**Near-optimal design scheme**  Unlike RL algorithm with limited switching cost, in multi-batch reinforcement learning, the agent can not change the policy adaptively. As a result, we need to design a policy with proper coverage ratio for all the survived policies. That is, using the data collected following this policy, the length of the confidence interval for any survived policy is bounded by a uniform threshold.

Recall that $d_h^\pi(s, a) = \mathbb{E}_\pi[\mathbb{I}[(s_h, a_h) = (s, a)]$. Using classical regret analysis for tabular RL [Azar et al., 2013, Zanette and Brunskill, 2019], for a fixed policy $\pi$, the length of confidence interval for $\pi$ could be roughly bounded by

$$\tilde{O}\left(\sum_{s,a,h} d_h^\pi(s, a)\sqrt{\frac{\text{Var}_h(s, a)}{N_h(s, a)}}\right) \underset{\text{Cauchy's ineq.}}{\leqslant} \tilde{O}\left(\sqrt{\sum_{s,a,h} \frac{d_h^\pi(s, a)}{N_h(s, a)}} \cdot \sqrt{\sum_{s,a,h} d_h^\pi(s, a)\text{Var}_h(s, a)}\right),$$
(3)

where $\text{Var}_h(s, a)$ is the variance term with respect to $P_{h,s,a}$ and $V*_{h+1}(\cdot)$, and $N_h(s, a) \geqslant 1$ is the count of $(h, s, a)$.

Because $\sum_{s,a,h} d_h^\pi(s, a)\text{Var}_h(s, a)$ could be uniformly bounded by $O(H^2)$ using classical analysis, we focus on bounding the term $\sum_{s,a,h} \frac{d_h^\pi(s,a)}{N_h(s,a)}$. Suppose the policy for current batch is $\tilde{\pi}$. After this batch, we roughly have that $N_h(s, a) \propto d_h^{\tilde{\pi}}(s, a)$. So it corresponds to find a policy $\tilde{\pi} \in \Pi(r, \mathcal{P})$ to minimize the *worst-case coverage number* $\max_{\pi \in \Pi(r, \mathcal{P})} \sum_{h,s,a} \frac{d_h^\pi(s,a)}{d_h^{\tilde{\pi}}(s,a)}$. For this problem, we have the lemma below, and the proof is deferred to Appendix E.1.

**Lemma 1.** *Let $d > 0$ be an integer. Let $\mathcal{X} \subset (\Delta^d)^m$. Then there exists a distribution $\mathcal{D}$ over $\mathcal{X}$, such that*

$$\max_{x=\{x_i\}_{i=1}^{dm} \in \mathcal{X}} \sum_{i=1}^{dm} \frac{x_i}{y_i} = md,$$

*where $y = \{y_i\}_{i=1}^{dm} = \mathbb{E}_{x\sim\mathcal{D}}[x]$. Moreover, if $\mathcal{X}$ has a boundary set $\partial\mathcal{X}$ with finite cardinality, we can find an approximation solution for $\mathcal{D}$ in $\text{poly}(|\partial\mathcal{X}|)$ time.*

Plugging $\mathcal{X} = \{\{d_h^\pi(\cdot, \cdot)\}_{h=1}^H | \pi \in \Pi(r, \mathcal{P})\}$, $d = SA$ and $m = H$ into Lemma 1, there exists a policy $\tilde{\pi}$ being a mixture of policies in $\Pi(r, \mathcal{P})$, such that $\max_{\pi \in \Pi(r, \mathcal{P})} \sum_{s,a,h} \frac{d_h^\pi(s,a)}{d_h^{\tilde{\pi}}(s,a)} = SAH$. In this way, we can find the desired exploration policy $\tilde{\pi}$ by assuming the knowledge of $\{d_h^\pi(\cdot, \cdot)\}_{h=1}^H$ for all $\pi \in \Pi(r, \mathcal{P})$.

Given the design scheme above, it remains two problems, for which we present solutions below: 1) $\{d_h^\pi(\cdot, \cdot)\}_{h=1}^H$ *is unknown;* 2) *even assuming $\{d_h^\pi(\cdot, \cdot)\}_{h=1}^H$ is known, it is hard to find $\tilde{\pi}$ since the cardinality of $\{\{d_h^\pi(\cdot, \cdot)\}_{h=1}^H | \pi \in \Pi(r, \mathcal{P})\}$ might be exponential in $SH$.*

**Constructing tight confidence region**  To estimate $\{d_h^\pi(\cdot, \cdot)\}_{h=1}^H$, we consider to construct a tight confidence region for the transition model to estimate the occupancy distribution up to a constant ratio.

**Definition 2.** *We say a confidence transition region $\mathcal{P} = \otimes_{h,s,a} \mathcal{P}_{h,s,a}$ is* tight *with respect to $p'$ iff (i) $p' \in \mathcal{P}$; (ii) $e^{-\frac{1}{H}} p'_{h,s,a,s'} \leqslant p_{h,s,a,s'} \leqslant e^{\frac{1}{H}} p'_{h,s,a,s'}$ for any $(h, s, a, s')$ and any $p_{h,s,a} \in \mathcal{P}_{h,s,a}$; (iii) $\mathcal{P}_{h,s,a}$ has the form $\mathcal{P}_{h,s,a} = \{p \in \Delta^S | a_i^\top p \leqslant b_i, i = 1, 2, ..., m\}$ where $m \leqslant \mathrm{poly}(SM)$.*

In model-based reinforcement learning, these conditions are natural and it is easy to construct a *tight* confidence region with acceptable error.

Once we have a confidence region which is *tight* w.r.t. the true transition model $P$, for any policy $\pi$ and $(h, s, a)$, we can estimate the expected visit count $W^\pi(\mathbf{1}_{h,s,a})$ by $W^\pi(\mathbf{1}_{h,s,a}, p)$ for any $p \in \mathcal{P}$ because

$$e^{-1} W^\pi(\mathbf{1}_{h,s,a}, p) \leqslant W^\pi(\mathbf{1}_{h,s,a}) = d_h^\pi(s, a) \leqslant e W^\pi(\mathbf{1}_{h,s,a}, p).$$

With $W^\pi(\mathbf{1}_{h,s,a}, p)$ as approximation of $d_h^\pi(s, a)$, we can continue the analysis above by paying a constant factor.

To learn such a confidence region, by Bennet's inequality (Lemma 3), it suffices to visit $(h, s, a, s')$[10] for $C_1 H^2 \iota$ for each $(h, s, a, s')$, where $C_1$ is an universal constant. By this idea, we try to visit each $(h, s, a, s')$ as much as possible. In the meantime, it is very possible that some $(h, s, a, s')$ tuples are extremely hard to visit. Fortunately, with proper exploration scheme, we can show that the maximal probability to visit such tuples is well-bounded, so that these tuples could be ignored by suffering regret $O(\sqrt{T})$.

**Computational efficient design scheme** Assume the confidence region $\mathcal{P}$ is *tight* w.r.t. $P$. We invoke reward-zero exploration to learn a sub-optimal solution for the problem $\min_{\tilde{\pi} \in \Pi(r, \mathcal{P})} \max_{\pi \in \Pi(r, \mathcal{P})} \sum_{h,s,a} \frac{d_h^\pi(s,a)}{d_h^{\tilde{\pi}}(s,a)}$. Let $p \in \mathcal{P}$ be fixed and define $\tilde{d}_h^\pi(s, a) = W^\pi(\mathbf{1}_{h,s,a}, p)$ be the approximation for $d_h^\pi(s, a)$. We define $\tilde{\pi}^i = \arg\max_{\pi \in \Pi(r, \mathcal{P})} W^\pi(r^i, p)$ for $1 \leqslant i \leqslant k = K^3$, where $r_h^i(s, a) = \min\left\{\frac{1}{\sum_{j=1}^{i-1} \tilde{d}_h^{\tilde{\pi}^j}(s,a)}, 1\right\}$. Let $\tilde{\pi}$ be the mixture of $\{\tilde{\pi}^i\}_{i=1}^k$. For any policy $\pi$, we have that

$$\sum_{s,a,h} d_h^\pi(s, a) \cdot \min\left\{\frac{1}{d_h^{\tilde{\pi}}(s,a)}, k\right\} \leqslant O\left(\sum_{s,a,h} \tilde{d}_h^\pi(s, a) \cdot \min\left\{\frac{1}{\tilde{d}_h^{\tilde{\pi}}(s,a)}, k\right\}\right) \tag{4}$$

$$\leqslant O\left(\sum_{i=1}^k W^\pi(r^i, p)\right) \tag{5}$$

$$\leqslant O\left(\sum_{i=1}^k W^{\tilde{\pi}^i}(r^i, p)\right) \tag{6}$$

$$\leqslant O\left(\sum_{s,a,h} \sum_{i=1}^k d_h^{\tilde{\pi}^i}(s, a) \cdot \min\left\{\frac{1}{\sum_{j=1}^{i-1} d_h^{\tilde{\pi}^j}(s,a)}, 1\right\}\right)$$

$$\leqslant O\left(\sum_{s,a,h} \sum_{i=1}^k \log\left(\frac{\max\{\sum_{j=1}^i d_h^{\tilde{\pi}^j}(s, a), 1\}}{\max\{\sum_{j=1}^{i-1} d_h^{\tilde{\pi}^j}(s, a), 1\}}\right)\right)$$

$$\leqslant O(SAH \log(k)). \tag{7}$$

Here (4) holds by the tightness of $\mathcal{P}$, (5) holds by the fact that $r_h^i(s, a) \geqslant r_h^{k+1}(s, a) = \min\left\{\frac{1}{\sum_{j=1}^k \tilde{d}_h^{\tilde{\pi}^j}(s,a)}, 1\right\} = \frac{1}{k} \min\left\{\frac{1}{\tilde{d}_h^{\tilde{\pi}}(s,a)}, k\right\}$ for any $(h, s, a)$, and (6) holds by the optimality of $\tilde{\pi}^i$ for $1 \leqslant i \leqslant k$. With (7) in hand, $\max_{\pi \in \Pi(r, \mathcal{P})} \sum_{h,s,a} \frac{d_h^\pi(s,a)}{d_h^\pi(s,a)}$ is roughly bounded by $O(SAH \log(K))$[11], which nearly matches the best *worst-case coverage number* number of $SAH$.

---

[10] A tuple $(h, s, a, s')$ is visited means $(s_h, a_h, s_{h+1}) = (s, a, s')$.

[11] We remark the there is still a gap between $\max_{\pi \in \Pi(r, \mathcal{P})} \sum_{h,s,a} \frac{d_h^\pi(s,a)}{d_h^\pi(s,a)}$ and $\sum_{s,a,h} d_h^\pi(s, a) \cdot \min\left\{\frac{1}{d_h^{\tilde{\pi}}(s,a)}, K^3\right\}$. Actually (7) is sufficient for further regret analysis.

---

**Algorithm 1** `Main Algorithm`

---

1: **Input:** state-action space $\mathcal{S} \times \mathcal{A}$, number of episodes $K$, confidence parameter $\delta$;
2: **Initialize:** $\iota \leftarrow \ln(2/\delta)$, $k_1 \leftarrow 144\sqrt{SAKH\iota}$, $k_2 \leftarrow 288S^3A^2H^4\sqrt{K\iota}$;
3: $\{\mathcal{D}_1\} \leftarrow$ `Raw Exploration`$(0, \varnothing, k_1)$;
4: $\{\mathcal{D}_2\} \leftarrow$ `Raw Exploration`$(r, \mathcal{D}_1, k_2)$;
5: `Policy Elimination`$(\mathcal{D}_2, K - Hk_1 - Hk_2)$.

---

**Computational efficient constrained exploration**  Let $u, u'$ be two reward functions and $\mathcal{P}$ be a set of transition models. As stated before, for general $\Pi(u, \mathcal{P})$, it might be non-trivial to solve the problem $\tilde{\pi} = \arg\max_{\pi \in \Pi(u,\mathcal{P})} W^\pi(u', p)$ for fixed $p \in \mathcal{P}$. As a trade-off, we turn to find some policy $\tilde{\pi} \in \Pi(u, \mathcal{P})$ such that $W^{\tilde{\pi}}(u', p) \geqslant c\max_{\pi \in \Pi(u,\mathcal{P})} W^\pi(u', p)$, where $c > 0$ is some universal constant. The problem turns out to be a RL problem with a soft constraint. For general $\Pi(u, \mathcal{P})$, the problem might be hard to solve. Fortunately, on the benefit of the *tight* property of $\mathcal{P}$, we can find such $\tilde{\pi}$ efficiently.

## 5 Algorithms

In this section we present our algorithms. The main algorithm (Algorithm 1) consists of three stages.

In the first two stages, we conduct naive exploration to identify the tuples which are hard to visit, which we called *infrequent* tuples. In particular, the length of the second stage is slightly larger than that of the first stage, where we use the dataset in the first stage to reduce the regret in the second stage. In this way, we can bound the regret in the first two stages by $\tilde{O}(\sqrt{SAH^3K})$, while the probability of visiting the *infrequent* tuples is small enough.

After ignoring the *infrequent* tuples, we could obtain a *tight* confidence region. Given the *tight* confidence region, we compute the confidence region for each policy and conduct policy elimination in the third stage. The first and second stages contains $O(H)$ batches, and the third stage contains $O(\log_2 \log_2(K))$ batches. So the batch complexity of Algorithm 1 is $O(H + \log_2 \log_2(K))$. Below we describe `Raw Exploration` (Algorithm 2) and `Policy Elimination` (Algorithm 3) in detail.

### 5.1 Raw Exploration

Given a dataset $\mathcal{D}$ with counts $\{N_h(s, a, s')\}$, we define the set of *known* tuples as $\{(h, s, a, s') : N_h(s, a, s') \geqslant C_1 H^2\iota\}$ and the left tuples are regarded as *infrequent* tuples.

In Algorithm 2, we are given a dataset. Then we compute the corresponding confidence region $\mathcal{P}$ in Line 20, where $\alpha(n, n') = \sqrt{\frac{4n'\iota}{n^2} + \frac{5\iota}{n}}$.

We conduct exploration layer by layer over policies in the set of survived policies $\Pi(r, \mathcal{P})$. By visiting each $(h, s, a)$ as much as possible, we can judge whether a tuple $(h, s, a, s')$ is hard to visit using policies in $\Pi(r, \mathcal{P})$.

Given the set of *known* tuples $\mathcal{W}$, we redirect all tuples not in $\mathcal{W}$ to an additional absorbed state $z$ using $\text{clip}(\cdot, \cdot)$. Once we prove that the probability of reaching $z$ is small enough for the any optimal policy, we can directly learn under the clipped transition model.

In Line 6 Algorithm 2, the algorithm `Policy Search` is invoked. Given any reward $u, u'$, any confidence region $\mathcal{P}$ and threshold $\epsilon > 0$, this algorithm returns a policy $\tilde{\pi} \in \Pi(u, \mathcal{P})$ such that $W^{\tilde{\pi}}(u', p) \geqslant c\max_{\pi \in \Pi(u,\mathcal{P})} W^\pi(u', p) - \epsilon$ with some universal constant $c > 0$. Moreover, when $\mathcal{P}$ is *tight* w.r.t. the true transition model $P$ after clipping, the time complexity of the algorithm is $O(\text{poly}(SAHK)\log(1/\epsilon))$. The algorithm and corresponding analysis is postponed to Appendix C.

It is also worth noting that executing each $\pi_{h,s,a}$ with probability $\frac{1}{SA}$ can not be regarded as a (history-independent) policy because the agent need to keep in mind which policy is chosen in current episode. In contrast, the agent only needs to observe current state to take actions following a policy. To address this problem, we define an operator `Sum` to take sum over policies under some transition model. Formally, we have the lemma below and postpone the proof to Appendix E.2.

**Algorithm 2** Raw Exploration$(u, \mathcal{D}, k)$

1: **Input**: reward function $u$, dataset $\mathcal{D}$, length $k$;
2: **Initialize:** $C_1 \leftarrow 200$;
3: **for** $h = 1, 2, \ldots, H$ **do**
4:     $\mathcal{P} \leftarrow \text{CR}(\mathcal{D})$;
5:     **for** $(s, a) \in \mathcal{S} \times \mathcal{A}$ **do**
6:         $\pi^{h,s,a} \leftarrow \text{Policy Search}(u, \mathbf{1}_{h,s,a}, \mathcal{P})$;
7:     **end for**
8:     $p \leftarrow$ arbitrary element in $\mathcal{P}$;
9:     $\{\tilde{\pi}^h, p\} \leftarrow \text{Sum}\left(\left\{\frac{1}{SA}, \pi^{h,s,a}, p\right\}_{(h,s,a)}\right)$;
10:    $\pi^h$ be the policy which is the same as $\tilde{\pi}^h$ in the first $h-1$ steps, and be the uniformly random policy in the left $H - h + 1$ steps;
11:    Execute $\pi^h$ for $k$ episodes, and collect the samples as $\mathcal{D}_h$;
12:    $\mathcal{D} \leftarrow \mathcal{D} \cup \mathcal{D}_h$;
13: **end for**
14: **return:** $\{\mathcal{D}\}$;

15: **Function**: CR$(\mathcal{D})$:
16:    $N_h(s, a, s') \leftarrow$ count of $(h, s, a, s')$ in $\mathcal{D}$, for all $(s, a, s')$;
17:    $N_h(s, a) \leftarrow \max\{\sum_{s'} N_h(s, a, s'), 1\}$ for all $(s, a)$;
18:    $\hat{p}_{h,s,a,s'} \leftarrow \frac{N_h(s,a,s')}{N_h(s,a)}, \forall (h, s, a, s')$;
19:    $\mathcal{W} \leftarrow \{(h, s, a, s') : N_h(s, a, s') \geqslant C_1 H^2 \iota\}$;
20:    $\tilde{\mathcal{P}}_{h,s,a} \leftarrow \{p \in \Delta^S | |p_{s'} - \hat{p}_{h,s,a,s'}| \leqslant \alpha(N_h(s, a), N_h(s, a, s')), \forall s' \in \mathcal{S}\}, \forall (h, s, a)$;
21:    $\mathcal{P}_{h,s,a} \leftarrow \{\text{clip}(p, \mathcal{W}) : p \in \tilde{\mathcal{P}}_{h,s,a}\}, \forall (h, s, a)$;
22:    **Return**: $\otimes_{h,s,a} \mathcal{P}_{h,s,a}$.

23: **Function**: clip$(p, \mathcal{W})$
24:    $p'_{h,s,a,s'} \leftarrow p_{h,s,a,s'}, \forall (h, s, a, s) \in \mathcal{W}$;
25:    $p'_{h,s,a,s'} \leftarrow 0, \forall (h, s, a, s') \notin \mathcal{W}$;
26:    $p'_{h,s,a,z} \leftarrow \sum_{s':(h,s,a,s') \notin \mathcal{W}} p_{h,s,a,s'}, \forall (h, s, a) \in [H] \times \mathcal{S} \times \mathcal{A}$;
27:    $p'_{h,z,a} \leftarrow \mathbf{1}_z, \forall (h, a) \in [H] \times \mathcal{A}$;
28:    **Return**: $p$.

---

**Lemma 2.** *Let $\mathcal{P} = \otimes_{(h,s,a)} \mathcal{P}_{h,s,a}$ be a set of transition models such that $\mathcal{P}_{h,s,a} \subset \Delta^S$ is convex for any $(h, s, a)$. Let $\{(\pi^i, P^i)\}_{i=1}^n$ be a sequence of policy-transition pairs such that $P^i \in \mathcal{P}$. For any $\{\lambda_i\}_{i=1}^n$ such that $\lambda_i \geqslant 0$ for $i \geqslant 1$ and $\sum_i \lambda_i = 1$, there exists a policy $\pi$ and $P \in \mathcal{P}$, satisfying that*

$$W^{\pi}(\mathbf{1}_{h,s,a}, P) = \sum_i \lambda_i W^{\pi^i}(\mathbf{1}_{h,s,a}, P^i) \tag{8}$$

*for any $(h, s, a) \in [H] \times \mathcal{S} \times \mathcal{A}$. Furthermore, the time complexity to find $\{\pi, P\}$ could be bounded by $O(nS^3 A^2 H^2)$.*

Therefore, for any $\{\lambda_i, \pi^i, P^i\}_{i=1}^n$ satisfying $\sum_{i=1}^n \lambda_i = 1$ and $\lambda_i \geqslant 0$ for $i \geqslant 1$ as input, there exists $\{\pi, P\}$ such that $W^{\pi}(\mathbf{1}_{h,s,a}, P) = \sum_i \lambda_i W^{\pi^i}(\mathbf{1}_{h,s,a}, P^i)$ and $P_{h,s,a} \in \text{Convex}(\{P^i_{h,s,a}\}_{i=1}^n)$ for any $(h, s, a) \in [H] \times \mathcal{S} \times \mathcal{A}$, where $\text{Convex}(\mathcal{U})$ denotes the convex hull of the set $\mathcal{U}$. Then Sum is defined as $\text{Sum}(\{\lambda_i, \pi^i, P^i\}_{i=1}^n) = \{\pi, P\}$.

## 5.2 Policy Elimination

Given the dataset collected in the first two stages, we first compute the *known* set $\mathcal{W}$. Unlike Algorithm 2, we do not update $\mathcal{W}$ in the rest time because the first two stages can ensure that the probability of visiting $\mathcal{W}^C$ is $O(1/\sqrt{K})$.

As mentioned in Section 4, for each batch, we invoke reward-zero exploration to search for the policy with near-optimal coverage. Based on such a policy, we can provide uniform bound for the length

---

**Algorithm 3** `Policy Elimination`

---

1: **Input:** dataset $\mathcal{D}$, length $k$;

2: **Initialize:** $\mathcal{D}^0 \leftarrow \mathcal{D}$, $\mathcal{P}^{-1} \leftarrow (\Delta^S)^{SA}$ $C_1 \leftarrow 100$, $v_h^{-1}(s) \leftarrow H - h + 1$, $\forall (h,s) \in [H] \times \mathcal{S}$;
$K_m \leftarrow \left\lceil K^{1-\frac{1}{2^m}} \right\rceil$ for $m = 1, 2, \ldots, M = \lceil \log_2 \log_2(K) \rceil$;

3: $N_h(s, a, s') \leftarrow$ count of $(h, s, a, s')$ in $\mathcal{D}$;

4: $\mathcal{W} \leftarrow \{(h, s, a, s') : N_h(s, a, s') \geqslant C_1 H^2 \iota\}$;

5: **for** $m = 0, 1, 2, \ldots, M - 1$ **do**

6: $\quad \mathcal{P}^m \leftarrow \mathcal{P}^{m-1} \cap \text{CR}^* \left( \mathcal{D}^m, \overline{\mathcal{D}}^m, \mathcal{W}, \{v_h^{m-1}(s)\}_{(h,s)} \right)$;

7: $\quad \pi^{m+1} \leftarrow \text{Design}((\mathcal{P}^m))$;

8: $\quad$ **if** $\sum_{m'=1}^{m} K_{m'} \leqslant k$ **then**

9: $\quad\quad$ Execute $\pi^{m+1}$ for $K_{m+1}$ episodes;

10: $\quad$ **else**

11: $\quad\quad$ Execute $\pi^{m+1}$ for $k - \left( \sum_{m'=1}^{m} K_{m'} \right)$ episodes;

12: $\quad$ **end if**

13: $\quad \overline{\mathcal{D}}^{m+1} \leftarrow$ the dataset in the $(m+1)$-th batch;

14: $\quad$ Update the dataset $\mathcal{D}^{m+1} \leftarrow \mathcal{D}^m \cup \overline{\mathcal{D}}^{m+1}$;

15: $\quad v_h^m(s) \leftarrow \max_{\pi, p \in \mathcal{P}^m} \mathbb{E}_{\pi, p} \left[ \sum_{h'=h}^{H} r_h(s_h, a_h) | s_h = s \right]$ for all $(h, s) \in [H] \times \mathcal{S}$;

16: **end for**

17: **Function**: $\text{CR}^*(\mathcal{D}, \mathcal{D}', \mathcal{W}, v)$:

18: $\quad \{N_h(s, a, s')\} \leftarrow$ counts in $\mathcal{D}$, $N_h(s, a) \leftarrow \max\{\sum_{s'} N_h(s, a, s'), 1\}$ for all $(h, s, a, s')$;

19: $\quad \hat{p}_{h,s,a,s'} \leftarrow \frac{N_h(s,a,s')}{N_h(s,a)}$, $\forall (h, s, a, s')$;

20: $\quad \{\check{N}_h(s, a, s')\} \leftarrow$ counts in $\mathcal{D}'$, $\check{N}_h(s, a) \leftarrow \max\{\sum_{s'} \check{N}_h(s, a, s'), 1\}$ for all $(h, s, a, s')$;

21: $\quad \check{p}_{h,s,a,s'} \leftarrow \frac{\check{N}_h(s,a,s')}{\check{N}_h(s,a)}$, $\forall (h, s, a, s')$;

22: $\quad \tilde{\mathcal{P}}_{h,s,a} \leftarrow \Big\{ p \in \Delta^S | |p_{s'} - \hat{p}_{h,s,a,s'}| \leqslant \alpha(N_h(s, a), N_h(s, a, s')), \forall s' \in \mathcal{S},$
$\quad\quad\quad\quad\quad\quad |(p - \check{p}_{h,s,a})v| \leqslant \alpha^*(\check{N}_h(s, a), \check{p}_{h,s,a}, v) \Big\}, \forall (h, s, a)$;

23: $\quad \mathcal{P}_{h,s,a} \leftarrow \{\text{clip}(p, \mathcal{W}) : p \in \tilde{\mathcal{P}}_{h,s,a}\}, \forall (h, s, a)$;

24: $\quad$ **Return**: $\otimes_{h,s,a} \mathcal{P}_{h,s,a}$.

25: **Function**: $\text{Design}(\mathcal{P})$:

26: $\quad p \leftarrow$ arbitrary element in $\mathcal{P}$;

27: $\quad$ **for** $i = 1, 2, ..., K^3$ **do**

28: $\quad\quad \tilde{d}_h^{\tilde{\pi}^j}(s, a) \leftarrow W^{\tilde{\pi}^j}(\mathbf{1}_{h,s,a}, p)$ for $1 \leqslant j \leqslant i - 1$ and any $(h, s, a)$;

29: $\quad\quad r_h^i(s, a) \leftarrow \min \left\{ \frac{1}{\sum_{j=1}^{i-1} \tilde{d}_h^{\tilde{\pi}^j}(s,a)}, 1 \right\}, \forall (h, s, a)$;

30: $\quad\quad \tilde{\pi}^i \leftarrow \text{Policy Search}(r, r^i, \mathcal{P})$;

31: $\quad$ **end for**

32: $\quad \{\pi, p\} \leftarrow \text{Sum} \left( \left\{ \frac{1}{K^3}, \tilde{\pi}^i, p \right\}_{i=1}^{K^3} \right)$;

33: $\quad$ **Return**: $\pi$.

---

of confidence intervals for all survived policies, which enables us to using the batch sizes in bandit algorithms [Perchet et al., 2016, Gao et al., 2019].

Besides, to obtain a better regret bound, we estimate the optimal value function at the end of each batch, and use it to build a tighter confidence region. As presented in Line 22 Algorithm 3, we use two empirical transition probabilities to construct the confidence region. Noting that the samples in the $m$-th batch is independent of $v^{m-1}$, we could add a Bernstein-style constraint, where

$$\alpha^*(n, p, v) = 5\sqrt{\frac{\mathbb{V}(p, v)\iota}{n}} + \frac{3\iota}{n}..$$

## 6 Conclusion

In this paper, we study multi-batch reinforcement learning, and provide an efficient algorithm to achieve the near-optimal regret bound and batch complexity. It would be an interesting problem to generalize our results to reinforcement learning with function approximation case, e.g., linear MDP. Another important direction is to study the exact batch-regret trade-off for multi-batch reinforcement learning.

**Broader Impact**  This work focus on the theory of multi-batch reinforcement learning, and the broader impact is not applicable.

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
