9: $\qquad\qquad$ Execute $\pi^{m+1}$ for $K_{m+1}$ episodes;
10: $\qquad$ **else**
11: $\qquad\qquad$ Execute $\pi^{m+1}$ for $k - (\sum_{m'=1}^{m} K_{m'})$ episodes;
12: $\qquad$ **end if**
13: $\qquad \overline{\mathcal{D}}^{m+1} \leftarrow$ the dataset in the $(m+1)$-th batch;
14: $\qquad$ Update the dataset $\mathcal{D}^{m+1} \leftarrow \mathcal{D}^m \cup \overline{\mathcal{D}}^{m+1}$;
15: $\qquad v_h^m(s) \leftarrow \max_{\pi, p \in \mathcal{P}^m} \mathbb{E}_{\pi,p}\left[\sum_{h'=h}^{H} r_h(s_h, a_h) | s_h = s\right]$ for all $(h,s) \in [H] \times \mathcal{S}$;
16: **end for**

17: **Function**: $\texttt{CR*}(\mathcal{D}, \mathcal{D}', \mathcal{W}, v)$:
18: $\qquad \{N_h(s,a,s')\} \leftarrow$ counts in $\mathcal{D}$, $N_h(s,a) \leftarrow \max\{\sum_{s'} N_h(s,a,s'), 1\}$ for all $(h,s,a,s')$;
19: $\qquad \hat{p}_{h,s,a,s'} \leftarrow \frac{N_h(s,a,s')}{N_h(s,a)}$, $\forall (h,s,a,s')$;
20: $\qquad \{\check{N}_h(s,a,s')\} \leftarrow$ counts in $\mathcal{D}'$, $\check{N}_h(s,a) \leftarrow \max\{\sum_{s'} \check{N}_h(s,a,s'), 1\}$ for all $(h,s,a,s')$;
21: $\qquad \check{p}_{h,s,a,s'} \leftarrow \frac{\check{N}_h(s,a,s')}{\check{N}_h(s,a)}$, $\forall (h,s,a,s')$;
22: $\qquad \tilde{\mathcal{P}}_{h,s,a} \leftarrow \Big\{ p \in \Delta^S | \, |p_{s'} - \hat{p}_{h,s,a,s'}| \leqslant \alpha(N_h(s,a), N_h(s,a,s')), \forall s' \in \mathcal{S},$
$\qquad\qquad\qquad\qquad\qquad |(p - \check{p}_{h,s,a})v| \leqslant \alpha^*(\check{N}_h(s,a), \check{p}_{h,s,a}, v) \Big\}, \forall (h,s,a)$;
23: $\qquad \mathcal{P}_{h,s,a} \leftarrow \{\texttt{clip}(p, \mathcal{W}) : p \in \tilde{\mathcal{P}}_{h,s,a}\}, \forall (h,s,a)$;
24: $\qquad$ **Return**: $\otimes_{h,s,a} \mathcal{P}_{h,s,a}$.

25: **Function**: $\texttt{Design}(\mathcal{P})$:
26: $\qquad p \leftarrow$ arbitrary element in $\mathcal{P}$;
27: $\qquad$ **for** $i = 1, 2, ..., K^3$ **do**
28: $\qquad\qquad \tilde{d}_h^{\tilde{\pi}^j}(s,a) \leftarrow W^{\tilde{\pi}^j}(\mathbf{1}_{h,s,a}, p)$ for $1 \leqslant j \leqslant i-1$ and any $(h,s,a)$;
29: $\qquad\qquad r_h^i(s,a) \leftarrow \min\left\{ \frac{1}{\sum_{j=1}^{i-1} \tilde{d}_h^{\tilde{\pi}^j}(s,a)}, 1\right\}, \forall (h,s,a)$;
30: $\qquad\qquad \tilde{\pi}^i \leftarrow \texttt{Policy Search}(r, r^i, \mathcal{P})$;
31: $\qquad$ **end for**
32: $\qquad \{\pi, p\} \leftarrow \texttt{Sum}\left(\left\{\frac{1}{K^3}, \tilde{\pi}^i, p\right\}_{i=1}^{K^3}\right)$;
33: $\qquad$ **Return**: $\pi$.

---

of confidence intervals for all survived policies, which enables us to using the batch sizes in bandit algorithms [Perchet et al., 2016, Gao et al., 2019].

Besides, to obtain a better regret bound, we estimate the optimal value function at the end of each batch, and use it to build a tighter confidence region. As presented in Line 22 Algorithm 3, we use two empirical transition probabilities to construct the confidence region. Noting that the samples in the $m$-th batch is independent of $v^{m-1}$, we could add a Bernstein-style constraint, where

$$\alpha^*(n, p, v) = 5\sqrt{\frac{\mathbb{V}(p,v)\iota}{n}} + \frac{3\iota}{n}..$$

## 6 Conclusion

In this paper, we study multi-batch reinforcement learning, and provide an efficient algorithm to achieve the near-optimal regret bound and batch complexity. It would be an interesting problem to generalize our results to reinforcement learning with function approximation case, e.g., linear MDP. Another important direction is to study the exact batch-regret trade-off for multi-batch reinforcement learning.

**Broader Impact**   This work focus on the theory of multi-batch reinforcement learning, and the broader impact is not applicable.

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

# A Technical Lemmas

**Lemma 3.** *Let $Z, Z_1, ..., Z_n$ be i.i.d. random variables with values in $[0, 1]$ and let $\delta > 0$. Define $\mathbb{V}Z = \mathbb{E}\left[(Z - \mathbb{E}Z)^2\right]$. Then we have*

$$\mathbb{P}\left[\left|\mathbb{E}\left[Z\right] - \frac{1}{n}\sum_{i=1}^{n} Z_i\right| > \sqrt{\frac{2\mathbb{V}Z\ln(2/\delta)}{n}} + \frac{\ln(2/\delta)}{n}\right] \leqslant \delta.$$

**Lemma 4.** *Let $X_1, X_2, \ldots$ be a sequence of random variables taking value in $[0, l]$. Define $\mathcal{F}_k = \sigma(X_1, X_2, \ldots, X_{k-1})$ and $Y_k = \mathbb{E}[X_k|\mathcal{F}_k]$ for $k \geqslant 1$. For any $\delta > 0$, we have that*

$$\mathbb{P}\left[\exists n, \sum_{k=1}^{n} X_k \leqslant 3\sum_{k=1}^{n} Y_k + l\ln(1/\delta)\right] \leqslant \delta$$

$$\mathbb{P}\left[\exists n, \sum_{k=1}^{n} Y_k \geqslant 3\sum_{k=1}^{n} X_k + l\ln(1/\delta)\right] \leqslant \delta.$$

*Proof.* Let $t \in [0, 1/l]$ be fixed. Consider to bound $Z_k := \mathbb{E}[\exp(t\sum_{k'=1}^{k}(X_{k'} - 3Y_{k'}))]$. By definition, we have that

$$\mathbb{E}[Z_k|\mathcal{F}_k] = \exp(t\sum_{k'=1}^{k}(X_{k'} - 3Y_{k'}))\mathbb{E}\left[t(X_k - 3Y_k)\right]$$

$$\leqslant \exp(t\sum_{k'=1}^{k}(X_{k'} - 3Y_{k'}))\exp(3Y_k) \cdot \mathbb{E}[1 + tX_k + 2t^2X_k^2]$$

$$\leqslant \exp(t\sum_{k'=1}^{k}(X_{k'} - 3Y_{k'}))\exp(3Y_k) \cdot \mathbb{E}[1 + 3tX_k]$$

$$= \exp(t\sum_{k'=1}^{k}(X_{k'} - 3Y_{k'}))\exp(3Y_k) \cdot (1 + 3tY_k)$$

$$\leqslant \exp(t\sum_{k'=1}^{k}(X_{k'} - 3Y_{k'}))$$

$$= Z_{k-1},$$

where the second line is by the fact that $e^x \leqslant 1 + x + 2x^2$ for $x \in [0, 1]$. Define $Z_0 = 1$ Then $\{Z_k\}_{k\geqslant 0}$ is a super-martingale with respect to $\{\mathcal{F}_k\}_{k\geqslant 1}$. Let $\tau$ be the smallest $n$ such that $\sum_{k=1}^{n} X_k - 3\sum_{k=1}^{n} Y_k > l\ln(1/\delta)$. It is easy to verify that $Z_{\min\{\tau, n\}} \leqslant \exp(tl\ln(1/\delta) + tl) < \infty$. Choose $t = 1/l$. By the optimal stopping time theorem, we have that

$$\mathbb{P}\left[\exists n \leqslant N, \sum_{k=1}^{n} X_k \geqslant 3\sum_{k=1}^{n} Y_k + l\ln(1/\delta)\right]$$

$$= \mathbb{P}\left[\tau \leqslant N\right]$$

$$\leqslant \mathbb{P}\left[Z_{\min\{\tau, N\}} \geqslant \exp(tl\ln(1/\delta))\right]$$

$$\leqslant \frac{\mathbb{E}[Z_{\min\{\tau, N\}}]}{\exp(tl\ln(1/\delta))}$$

$$\leqslant \delta.$$

Letting $N \to \infty$, we have that

$$\mathbb{P}\left[\exists n, \sum_{k=1}^{n} X_k \leqslant 3\sum_{k=1}^{n} Y_k + l\ln(1/\delta)\right] \leqslant \delta.$$

Considering $W_k = \mathbb{E}[\exp(t \sum_{k'=1}^{k}(Y_k/3 - X_k))]$, using similar arguments and choosing $t = 1/(3l)$, we have that

$$\mathbb{P}\left[\exists n, \sum_{k=1}^{n} Y_k \geqslant 3\sum_{k=1}^{n} X_k + l\ln(1/\delta)\right] \leqslant \delta.$$

The proof is completed. $\qquad\square$

**Lemma 5.** *Let the policy $\pi$ and reward $r$ be fixed. Let $p$ and $p'$ be two transition model, it holds that*

$$W^\pi(r, p) - W^\pi(r, p') = \sum_{h,s,a} W^\pi(\mathbf{1}_{h,s,a}, p)(p'_{h,s,a} - p_{h,s,a})V'_{h+1}, \tag{9}$$

*where $\{V'_h(s)\}_{(h,s)\in[H]\times\mathcal{S}}$ is the value function under $p'$ following $\pi$.*

## B  Lower Bound (Proof of Theorem 2)

Firstly, by the lower bound on batched bandit (Theorem 3 in [Gao et al., 2019]), to achieve $O(\text{poly}(S, A, H)\sqrt{K})$ regret, the number of batches is at least $\Omega(\log_2\log_2(K))$. To show a lower bound of $\Omega(H/\log_A(K))$, we have the lemma below by considering an MDP with 2 states and $A$ actions.

**Lemma 6.** *Let $\mathcal{S} = \{s^{(0)}, s^{(1)}\}$, $\mathcal{A} = \{a_0, a_1, \ldots, a_A\}$ and $s_1 = s^{(0)}$. Let $d = \lfloor 2\log_A(K)\rfloor + 2$. For $v = [v_1, v_2, \ldots, v_d]^\top \in A^d$, we define the transition model $P^v$ by setting $P^v_{h,s^{(0)},a_x} = [1,0]^\top, \forall x \neq v_h$, $P^v_{h,s^{(0)},a_{v_h}} = [0,1]^\top$ and $P^v_{h,s^{(1)},a_x} = [0,1]^\top, \forall 1 \leqslant x \leqslant A$ for $1 \leqslant h \leqslant d$. Let $\pi$ be a stochastic policy, Then there exists $v$ such that with probability $1 - \frac{1}{K}$, $(h, s^{(0)})$ is never visited in $K$ episodes following $\pi$.*

*Proof.* Denote the distribution of $\pi$ as $\mathsf{D}$, we define $x = [x_1, x_2, \ldots, x_d]^\top$ as below. Let $x_1 = \arg\min_i \mathbb{E}_{\pi\sim D}\left[\pi_1(a_i|s^{(0)})\right]$. For $2 \leqslant h \leqslant d$, we define

$$x_h = \arg\max_i \frac{\mathbb{E}_{\pi\sim\mathsf{D}}\left[\mathbb{I}_\pi[s_{h-1} = s^{(0)}|P^{x,h-1}]\pi_h(a_i|s_h)\right]}{\mathbb{E}_{\pi\sim\mathsf{D}}\left[\mathbb{I}_\pi[s_{h-1} = s^{(0)}|P^{x,h-1}]\right]},$$

where $P^{x,h-1}$ denote the first $(h-1)$-layers of the transition model $P^x$. Because $\mathbb{P}_\pi[s_h = s^{(0)}|P^x]$ is determined by the first $(h-1)$-layers of $P^x$, $x_h$ is well-defined. By definition we have that

$$\frac{\mathbb{E}_{\pi\sim\mathsf{D}}\left[\mathbb{I}_\pi[s_{h-1} = s^{(0)}|P^{x,h-1}]\pi_h(a_{x_h}|s_h)\right]}{\mathbb{E}_{\pi\sim\mathsf{D}}\left[\mathbb{I}_\pi[s_{h-1} = s^{(0)}|P^{x,h-1}]\right]} \leqslant \frac{1}{A}. \tag{10}$$

Recall that $x = [x_1, x_2, \ldots, x_d]^\top$. For $1 \leqslant h' \leqslant d$, by (10) we have that

$$\mathbb{E}_{\pi\sim\mathsf{D}}\mathbb{P}_\pi\left[s_{h'} = s^{(0)}|P^x\right]$$
$$= \mathbb{E}_{\pi\sim\mathsf{D}}\Pi_{h=1}^{h'-1}\pi_h(a_{x_h}|s^{(0)})$$
$$= \mathbb{E}_{\pi\sim\mathsf{D}}\mathbb{P}_\pi\left[s_{h'-1} = s^{(0)}|P^x\right] \cdot \frac{\mathbb{E}_{\pi\sim\mathsf{D}}\left[\mathbb{I}_\pi[s_{h'-1} = s^{(0)}|P^{x,h-1}]\pi_h(a_{x_h}|s_h)\right]}{\mathbb{E}_{\pi\sim\mathsf{D}}\left[\mathbb{I}_\pi[s_{h'-1} = s^{(0)}|P^{x,h-1}]\right]}$$
$$\leqslant \frac{1}{A}\mathbb{E}_{\pi\sim\mathsf{D}}\mathbb{P}_\pi\left[s_{h'-1} = s^{(0)}|P^x\right]. \tag{11}$$

Therefore, $\mathbb{E}_{\pi\sim\mathsf{D}}\mathbb{P}_\pi\left[s_d = s^{(0)}|P^x\right] \leqslant \frac{1}{A^{d-1}} \leqslant \frac{1}{K}$. Then the probability of visiting $(h, s^{(0)})$ in $K$ episodes is at most $\frac{1}{K}$, where the conclusion follows. $\qquad\square$

We name the MDP in Lemma 6 as a basic MDP. Now we construct our counter-example by concatenating $\Theta(H/\log_A(K))$ basic MDPs and a tail MDP with large rewards. Let $\mathcal{S} = \{s^{(0)}, s^{(1)}\}$ and $\mathcal{A} = \{a_1, a_2, \ldots, a_A\}$. Let $d = \lfloor 2\log_A(K)\rfloor + 2$ and $c = \lfloor\frac{H}{2d}\rfloor$. Then $c = C'H/\log_A(K)$ for some constant $C'$. For $v = [v_1, v_2, \ldots, v_{cd}]^\top \in \{0,1\}^{cd}$, we define the transition model $P^v$ as below: $P^v_{id+j,s^{(0)},a_{v_{id+j}}} = [1,0]^\top$, $P^v_{id+j,s^{(0)},a_l} = [0,1]^T$ for $l \neq v_{id+j}$ and $P^v_{id+j,s^{(1)},a_l} = [0,1]^\top$

for $1 \leqslant l \leqslant A$ for any $0 \leqslant i \leqslant c-1$ and $1 \leqslant j \leqslant d$; $P_{h,s^{(0)},a_l} = [1,0]^\top$ and $P_{h,s^{(1)},a_l} = [0,1]^\top$ for any $1 \leqslant l \leqslant A$ and $cd+1 \leqslant h \leqslant H$. The reward function $r$ is given by $r_{h,s^{(0)},a_l}$ for $1 \leqslant l \leqslant A$ $cd+1 \leqslant h \leqslant H$ and $0$ for other $(h,s,a)$ triples.

To achieve sub-linear regret, the agent needs to visit $(cd+1, s^{(0)})$ for at least one time. Then the proof is completed by the lemma below.

**Lemma 7.** *If the number of batches $M \leqslant c-2$, for any algorithm $\mathcal{G}$ there exists $v$ such that with probability $1 - \frac{c}{K} \geqslant \frac{1}{2}$, $(cd+1, s^{(0)})$ is never visited.*

*Proof.* Let $m_i$ denote the number of batches used at the time when $(id+1, s^{(0)})$ is visited for the first time. Besides, we let $\pi(i)$ denote the policy at time $m_i$. Because $\pi(i)$ is determined before visiting $(id+1, s^{(0)})$, given the algorithm $\mathcal{G}$, $\pi^i$ could be viewed as a stochastic function of $\{v_1, v_2, \ldots, v_{id}\}$. By Lemma 6, when $\{v_1, v_2, \ldots, v_{id}\}$ is fixed, we can choose $\{v_{id+1}, \ldots, v_{id+d}\}$ properly, so that with probability $1 - \frac{1}{K}$, $((i+1)d+1, s^{(0)})$ is never visited in $K$ episodes following $\pi(i)$. Therefore, with probability $1 - \frac{1}{K}$, $\pi(i+1) \neq \pi(i)$, which implies that $m_{i+1} \geqslant m_i + 1$. By choosing $\{v_{id+1}, v_{id+1}, \ldots, v_{id+d}\}$ recursively following the way in Lemma 6 for $0 \leqslant i \leqslant c-1$, we have that with probability $1 - \frac{c}{K}$, $m_{i+1} \geqslant m_i + 1$ for $1 \leqslant i \leqslant c$, where $m_c \geqslant c-1$ follows. Then the conclusion follows by the equation below.

$$\mathbb{P}\left[M \leqslant c-2, \ (cd+1, s^{(0)}) \text{ is visited}\right] = \mathbb{P}\left[m_c \leqslant c-2\right] \leqslant \frac{c}{M}.$$

$\square$

## C  Efficient Implementation of the Proposed Algorithm

In this section, we analyze the computational cost of Algorithm 1. In particular, we first introduce the algorithm `PolicySearch` to show that it can help find the desired exploration policy efficiently.

### C.1  The Algorithm

`Policy Search` is presented in Algorithm 4. The algorithms takes two reward functions $u, u'$ and a confidence region $\mathcal{P}$ as input, and output a policy $\pi$ and $p \in \mathcal{P}$ such that $W^\pi(u', P)$ is large enough compared to $\max_{\pi' \in \Pi(u,\mathcal{P}),} W^{\pi'}(u', P)$.

In the algorithm, we first compute $a := \max_\pi U^\pi(u + \mathbf{1}_z, \mathcal{P})$ and $b := \max_\pi L^\pi(u, \mathcal{P})$. Then we set the target reward as $u + \mathbf{1}_z + \eta u'$ for different $\eta$ and learn the corresponding optimal policy and transition model $\{\pi^\eta, P^\eta\}$. In intuition, the larger $\eta$ is, the larger $W^{\pi^\eta}(u, P^\eta)$ is. In this way, we aim to find the maximal $\eta$ such that $\pi^\eta$ is not eliminated, i.e., $\pi^\eta \in \Pi(u, \mathcal{P})$. To find such $\eta$, we play the naive dichotomy method as presented in Algorithm 4.

When $u = r$, we assume that $a - b \geqslant \frac{1}{K^3}$ without loss of generality. Note that when $a - b \leqslant \frac{1}{K^3}$, any policy $\pi$ in $\Pi(r, \mathcal{P})$ is $\frac{1}{K^3}$ optimal and we can follow $\pi$ in the rest episodes.

In Algorithm 4, we invoke extended value iteration (EVI, see Algorithm 5) as a sub-routine. Algorithm 5 targets compute $(\pi, p) \leftarrow \arg\max_{\pi, p \in \mathcal{P}} W^\pi(u, p)$ for some reward function $u$ and confidence region $\mathcal{P}$. In finite-horizon MDP, this step could be implemented by back induction. So it suffices to solve $\arg\max_{a, p \in \mathcal{P}} p_{h,s,a} V_{h+1}$ where $V_{h+1}$ is the value function computed by back induction. Note that in this paper, the confidence region could be described by at most $O(S^2 AK)$ linear constraints, which enables us to find an approximate solution in polynomial time. Besides, given $u$ and $\mathcal{P}$, $\max \pi U^\pi(u, \mathcal{P})$ and $\max_\pi L^\pi(u, \mathcal{P})$ could be computed in a similar way, for which we present Algorithm 6. As a conclusion, Algorithm 4 is computationally efficient.

### C.2  Theoretical Results and Proofs for Algorithm 4

**Lemma 8.** *Let $u, u'$ be two reward functions and $\mathcal{P}$ be a set of transition models. Assume $\mathcal{P} = \otimes_{h,s,a} \mathcal{P}_{h,s,a}$ is tight w.r.t. a transition model $P$. Then by Algorithm 4 we can find $\pi$ such that*

$$W^\pi(u', P) \geqslant \frac{1}{18} \max_{\pi' \in \Pi(u, \mathcal{P}),} W^{\pi'}(u', P) - \frac{2}{9}\epsilon$$

---

**Algorithm 4** `Policy Search`

---

**Input:** reward $u$, $u'$, confidence region $\mathcal{P} = \otimes_{h,s,a} \mathcal{P}_{h,s,a}$;

**Initialization:** threshold $\epsilon = \frac{1}{(SAHK)^{10}}$, $b \leftarrow \max_\pi L^\pi(u, \mathcal{P})$, $a \leftarrow \max_\pi U^\pi(u + \mathbf{1}_z, \mathcal{P})$;

$\eta_0 \leftarrow (a - b)/2$;

**for** $i = 0, 1, 2, \ldots,$ **do**

    $\{\pi^{(i)}, P^{(i)}\} \leftarrow \texttt{EVI}(u + \mathbf{1}_z + \eta_i u', \mathcal{P})$;

    **if** $\frac{1}{\epsilon} \leqslant \eta_i < \frac{2}{\epsilon}$; **then**

      **Return:** $\pi^{(i)}$;

    **else if** $W^{\pi^{(i)}}(u, P^{(i)}) \leqslant b$ **then**

      $\xi = \frac{b - W^{\pi^{(i)}}(u, P^{(i)})}{W^{\pi^{(i-1)}}(u, P^{(i-1)}) - W^{\pi^{(i)}}(u, P^{(i)})}$;

      $(\check{\pi}, \check{P}) \leftarrow \texttt{Sum}(\{\xi, \pi^{(i-1)}, P^{(i-1)}\}, \{1 - \xi, \pi^{(i)}, P^{(i)}\})$

      **Return:** $\check{\pi}$ ;

    **else**

      $\eta_{i+1} = 2\eta_i$;

    **end if**

**end for**

---

 

---

**Algorithm 5** `Extended Value Iteration (EVI)`

---

**Input:** reward function $u$, confidence region $\mathcal{P} = \otimes_{h,s,a} \mathcal{P}_{h,s,a}$

**Initialize:** $Q_h(s, a) \leftarrow 0, V_h(s) \leftarrow 0, \forall (h, s, a) \in [H + 1] \times \mathcal{S} \times \mathcal{A}$

**for** $h = H, H - 1, \ldots, 1$ **do**

    $Q_h(s, a) \leftarrow \max_{q \in \mathcal{P}_{h,s,a}} (u(s, a) + q V_{h+1}), \forall (s, a) \in \mathcal{S} \times \mathcal{A}$;

    $p_{h,s,a} \leftarrow \arg\max_{q \in \mathcal{P}_{h,s,a}} (u(s, a) + q V_{h+1})$;

    $V_h(s) \leftarrow \max_a Q_h(s, a), \forall s \in \mathcal{S}$;

    $\pi_h(a|s) \leftarrow \mathbb{I}[a = \arg\max_{a'} Q_h(s, a')], \forall (s, a)$;

**end for**

**Return:** $\{\pi, p\}$.

---

 

---

**Algorithm 6** `Upper&Lower Confidence Bound`

---

**Input:** reward function $u$, confidence region $\mathcal{P} = \otimes_{h,s,a} \mathcal{P}_{h,s,a}$;

**Initialize:** $\overline{Q}_h(s, a), \overline{V}_h(s), \underline{Q}_h(s, a), \underline{V}_h(s) \leftarrow 0, \forall (h, s, a) \in [H + 1] \times \mathcal{S} \times \mathcal{A}$;

**for** $h = H, H - 1, \ldots, 1$ **do**

    $\overline{Q}_h(s, a) \leftarrow \max_{q \in \mathcal{P}_{h,s,a}} (u(s, a) + q \overline{V}_{h+1}), \forall (s, a) \in \mathcal{S} \times \mathcal{A}$;

    $\overline{V}_h(s) \leftarrow \max_a \overline{Q}_h(s, a), \forall s \in \mathcal{S}$;

    $\underline{Q}_h(s, a) \leftarrow \min_{q \in \mathcal{P}_{h,s,a}} (u(s, a) + q \underline{V}_{h+1}), \forall (s, a) \in \mathcal{S} \times \mathcal{A}$;

    $\underline{V}_h(s) \leftarrow \max_a \underline{Q}_h(s, a), \forall s \in \mathcal{S}$;

**end for**

**Return:** $\max_\pi U^\pi(u, \mathcal{P}) := \overline{V}_1(s_1), \max_\pi L^\pi(u, \mathcal{P}) := \underline{V}_1(s_1)$;

---

*in time* $O(S^4 AHM^3 \log(SAHK) \log(SAHK/(a-b)))$, *where* $a = \max_\pi U^\pi(u + \mathbf{1}_z, \mathcal{P})$ *and* $b = \max_\pi L^\pi(u, \mathcal{P})$.

*Proof.* Let $\tilde{u} = u + \mathbf{1}_z$. For any $\eta \geqslant 0$, we define $(\pi^\eta, p^\eta)$ be the policy-transition pair such that

$$(\pi^\eta, P^\eta) = \arg \max_{\pi, p \in \mathcal{P}} W^\pi(\tilde{u} + \eta u', p).$$

By Lemma 10, with Algorithm 6, we can compute $a$ and $b$ within time $\tilde{O}\left(S^4 AHM^3 \log(SAHK)\right)$. In the same way, with Algorithm 5 we can find $(\pi^\eta, p^\eta)$ within time $\tilde{O}\left(S^4 AHM^3 \log(SAHK)\right)$ for any $\eta > 0$. Note that in Algorithm 4, the value of $i$ is at most $\log(1/(\eta_0 \epsilon)) = O(\log(\frac{1}{\epsilon(a-b)})) = O(\log(SAHK))$. As a result, the computational cost is at most $O(S^4 AHM^3 \log(SAHK) \log(SAHK/(a-b)))$.

We continue with an useful property of $(\pi^\eta, P^\eta)$.

**Lemma 9.** *Let* $0 < \eta < \eta'$ *be fixed. Let* $(\pi^\eta, p^\eta)$, $(\pi^{\eta'}, P^{\eta'})$ *be such that*

$$(\pi^\eta, P^\eta) = \arg \max_{\pi, p \in \mathcal{P}} W^\pi(\tilde{u} + \eta u', p)$$

$$(\pi^{\eta'}, P^{\eta'}) = \arg \max_{\pi, p \in \mathcal{P}} W^\pi(\tilde{u} + \eta' u', p).$$

*Then we have that*

$$W^{\pi^\eta}(\tilde{u}, P^\eta) \geqslant W^{\pi^{\eta'}}(\tilde{u}, P^{\eta'}).$$

*Proof.* Let $x_1 = W^{\pi^\eta}(\tilde{u}, P^\eta)$, $x_2 = W^{\pi^{\eta'}}(\tilde{u}, P^{\eta'})$, $y_1 = W^{\pi^\eta}(u', P^\eta)$ and $y_2 = W^{\pi^{\eta'}}(u', P^{\eta'})$. It suffices to show that $x_1 \geqslant x_2$. By the optimality of $(\pi^\eta, P^\eta)$ and $(\pi^{\eta'}, P^{\eta'})$, we have that

$$x_1 + \eta y_1 \geqslant x_2 + \eta y_2;$$
$$x_2 + \eta' y_2 \geqslant x_1 + \eta' y_1.$$

If $x_1 < x_2$, then we have that $y_1 > y_2$. It then follows that $x_2 + \eta' y_2 = x_2 + \eta y_2 + (\eta' - \eta) y_2 < x_1 + \eta y_1 + (\eta' - \eta) y_1 = x_1 + \eta' y_1$, which leads to contradiction. □

In Algorithm 4, there are two breaking conditions.

**Case 1** Recall that $\{\pi^{(i)}, P^{(i)}\} = \arg \max_{\pi, p \in \mathcal{P}} W^\pi(u + \mathbf{1}_z + \eta_i \mu', p) = \arg \max_{\pi, p \in \mathcal{P}} W^\pi(\tilde{u} + \eta_i \mu', p)$ In the first case, we end with obtaining some $i$ satisfying that

$$W^{\pi^{(i)}}(\tilde{u}, P^{(i)}) \leqslant b.$$

Because $W^{\pi^{(0)}}(\tilde{u}, P^{(0)}) \geqslant a - \eta_0 > b$, it holds that $\eta_i > \eta_0$ for any $i \geqslant 1$. By Lemma 9 and the stopping condition, we have that $W^{\pi^{(i-1)}}(\tilde{u}, P^{(i-1)}) \geqslant b$. By Lemma 2, we can find a policy $\check{\pi}$ and $\check{P} \in \mathcal{P}$ such that

$$W^{\check{\pi}}(v, \check{P}) = \xi W^{\pi^{(i)}}(v, P^{(i)}) + (1 - \xi) W^{\pi^{(i-1)}}(v, P^{(i-1)}) \tag{12}$$

for any reward function $v$.

Noting that $\xi = \frac{b - W^{\pi^{(i)}}(\tilde{u}, P^{(i)})}{W^{\pi^{(i-1)}}(\tilde{u}, P^{(i-1)}) - W^{\pi^{(i)}}(\tilde{u}, P^{(i)})}$, we have that $U^{\check{\pi}}(u, \mathcal{P}) \geqslant W^{\check{\pi}}(\tilde{u}, \check{P}) = \xi W^{\pi^{(i)}}(\tilde{u}, P^{(i)}) + (1 - \xi) W^{\pi^{(i-1)}}(\tilde{u}, P^{(i-1)}) = b$, which implies that $\check{\pi} \in \Pi(u, \mathcal{P})$.

Note that $W^\pi(v, p)$ is linear in $v$ for fixed $\pi$ and $p$. For any policy $\pi \in \Pi(r, \mathcal{P})$ and $p' \in \mathcal{P}$, we have that

$$W^\pi(\tilde{u}, p') + \eta_i W^\pi(u', p') \leqslant W^{\pi^{(i)}}(\tilde{u}, p^{(i)}) + \eta_i W^{\pi^{(i)}}(u', P^{(i)}), \tag{13}$$

$$W^\pi(\tilde{u}, p') + \eta_{i-1} W^\pi(u', p') \leqslant W^{\pi^{(i-1)}}(\tilde{u}, P^{(i-1)}) + \eta_{i-1} W^{\pi^{(i-1)}}(u', P^{(i-1)}). \tag{14}$$

It then follows that

$$W^\pi(\tilde{u}, p') + \eta_{i-1} W^\pi(u', p')$$

$$\leqslant \xi \left( W^{\pi^{(i)}}(\tilde{u}, P^{(i)}) + \eta_i W^{\pi^{(i)}}(u', P^{(i)}) \right) + (1 - \xi) \left( W^{\pi^{(i-1)}}(\tilde{u}, P^{(i-1)}) + \eta_{i-1} W^{\pi^{(i-1)}}(u', P^{(i-1)}) \right)$$

$$\leqslant b + \eta_i W^{\check{\pi}}(u', \check{P}). \tag{15}$$

For any $\pi \in \Pi(u, \mathcal{P})$, there exists $p' \in \Pi(u, \mathcal{P})$ such that $W^\pi(\tilde{u}, p') \geqslant b$. By (15) and noting that $\eta_i = 2\eta_{i-1}$, we have

$$W^\pi(u', p') \leqslant \frac{\eta_i}{\eta_{i-1}} W^{\check{\pi}}(u', \check{P}) \leqslant 2 W^{\check{\pi}}(u', \check{P}). \tag{16}$$

On the other hand, by Lemma 17, for any $\pi$ it holds that

$$W^\pi(u', p) \leqslant 3 W^\pi(u', p') \leqslant 9 W^\pi(u', p), \tag{17}$$

for any $p' \in \bar{\mathcal{P}}$, which implies that

$$W^{\check{\pi}}(u', p) \geqslant \frac{1}{6} \max_{\pi \in \Pi(u, \mathcal{P})} W^\pi(u', p).$$

**Case 2**  In the second case, we end with some $i$ such that $\frac{1}{\epsilon} \leqslant \eta_i < \frac{2}{\epsilon}$.

In this case, because $W^{\pi^{(i)}}(\tilde{u}, P^{(i)}) \geqslant b$, we have that $\pi^{(i)} \in \Pi(u, \mathcal{P})$ . For any $\pi \in \Pi(u, \mathcal{P})$ such that

$$W^\pi(u', p) \geqslant 18 W^{\pi^{(i)}}(u', p), \tag{18}$$

by the *tightness* of $\mathcal{P}$ (w.r.t. $p$) it holds that

$$\eta_i W^\pi(u', p') \geqslant \frac{\eta_i}{3} W^\pi(u', p) \geqslant 6\eta_i W^{\pi^{(i)}}(u', p) \geqslant 2\eta_i W^{\pi^{(i)}}(u', P^{(i)}) \tag{19}$$

for any $p' \in \mathcal{P}$. On the other hand, by optimality of $(\pi^{(i)}, P^{(i)})$, we have that

$$\eta_i W^\pi(u', p') \leqslant W^{\pi^{(i)}}(\tilde{u}, P^{(i)}) + \eta_i W^{\pi^{(i)}}(u', P^{(i)}). \tag{20}$$

Combine (19) with (20), we have that

$$\eta_i W^{\pi^{(i)}}(u', P^{(i)}) \leqslant W^{\pi^{(i)}}(\tilde{u}, P^{(i)}) \leqslant 2. \tag{21}$$

Combining (20) with (21), for any $p' \in \mathcal{P}$, using the optimality of $(\pi^{(i)}, P^{(i)})$ and (21), we have that

$$\eta_i W^\pi(u', p') \leqslant W^{\pi^{(i)}}(u, P^{(i)}) + \eta_i W^{\pi^{(i)}}(u', P^{(i)}) \leqslant 4. \tag{22}$$

It then follows $W^\pi(u', p) \leqslant 4\epsilon$. Therefore, for any $\pi \in \Pi(u, \mathcal{P})$, it holds either $W^\pi(u', p) \leqslant 18 W^{\pi^{(i)}}(u', p)$ or $W^\pi(u', p) \leqslant 4\epsilon$. We then have that

$$W^{\pi^{(i)}}(u', p) \geqslant \frac{1}{18} \max_{\pi \in \Pi(u, \mathcal{P})} W^\pi(u', p) - \frac{2}{9}\epsilon. \tag{23}$$

The proof is completed.

$\square$

**Lemma 10.** *The computational cost of Algorithm 5 and Algorithm 6 is bounded by* $O(S^3 AHM^3 \log(SAKH))$.

*Proof.* To implement the two algorithm, we need to solve $SAH$ linear optimization problem, which has the form $\max_{q \in \mathcal{P}_{h,s,a}}(r + qv)$ or $\min_{q \in \mathcal{P}_{h,s,a}}(r + qv)$. Note that $\mathcal{P}_{h,s,a}$ has the form $\{p \in \Delta^{\mathcal{S}} : a_i^\top(p - p') \leqslant b_i, i \geqslant 1\}$, and the number of linear constraints is increased for at most $O(S)$ in each batch. As a result, the total number of linear constraints in $\mathcal{P}_{h,s,a}$ is bounded by $O(SM)$. By the results in Cohen et al. [2021], the time cost to solve the linear program problem above is bounded by $O(S^3 M^3 \log(SAHK))$. Therefore, the total computational cost is bounded by $O(S^3 AHM^3 \log(SAKH))$. $\square$

# D  Proof of Theorem 1

**Additional Notations**  In this section, we use $N_h^m(s, a, s')$ to denote the visit count of $(s, a, h, s')$ after the $m$-th batch. We also define $N_h^m(s, a) = \max\{\sum_{s'} N_h^m(s, a, s'), 1\}$. We use $\{\check{N}_h^m(s, a, s')\}$ to denote the counts of the $m$-th batch. Similarly we define $\check{N}_h^m(s, a) = \max\{\sum_{s'} \check{N}_h^m(s, a, s'), 1\}$. Let $W^*$ be the *known* set after the first two stages. Let $\hat{P}_{h,s,a,s'}^m = \frac{N_h^m(s,a,s')}{N_h^m(s,a)}$ be the empirical transition model for $1 \leqslant m \leqslant 2H + M$. For $2H + 1 \leqslant m \leqslant 2H + M$, define $\{\check{P}_{h,s,a}^m\}$ be the clipped transition model, i.e., $\{\check{P}_{h,s,a}^m\}_{h,s,a} = \texttt{clip}\left(\left\{\left[\frac{\check{N}_h^m(s,a,s')}{\check{N}_h^m(s,a)}\right]_{s'\in\mathcal{S}}\right\}_{h,s,a}, \mathcal{W}^*\right)$.

Note that the $m$-batch in Algorithm 3 indicates the $2H + m$-th batch in the main algorithm. To align the indices, with a slight abuse of notations we use $\mathcal{P}^m$ and $v^m$ to denote respectively the value of $\mathcal{P}^{m-2H}$ and $v^{m-2H}$ in Algorithm 3 for $m \geqslant 2H$.

Table 1: Explanation of the notations

| | |
|---|---|
| $W^\pi(u, p)$ | the general value function: $W^\pi(u, p) = \mathbb{E}_{p,\pi,s_1\sim\mu_1}[\sum_{h=1}^H u_h(s_h, a_h)]$ |
| $U^\pi(u, \mathcal{P})$ | the upper confidence bound w.r.t. policy $\pi$, reward $u$ and confidence region $\mathcal{P}$ ; |
| $L^\pi(u, \mathcal{P})$ | the lower confidence bound w.r.t. policy $\pi$, reward $u$ and confidence region $\mathcal{P}$ ; |
| $N_h^m(s, a, s')$ | the visit count of $(s, a, h, s')$ after the $m$-th batch |
| $N_h^m(s, a)$ | $N_h^m(s, a) = \max\{\sum_{s'} N_h^m(s, a, s'), 1\}$; |
| $\check{N}_h^m(s, a, s')$ | the count of $(h, s, a, s')$ in the $m$-th batch; |
| $\check{N}_h^m(s, a)$ | $\check{N}_h^m(s, a) = \max\{\sum_{s'} \check{N}_h^m(s, a, s'), 1\}$ |
| $W^*$ | the *known* set after the first two stages |
| $\hat{P}_{h,s,a,s'}^m$ | $\hat{P}_{h,s,a,s'}^m = \frac{N_h^m(s,a,s')}{N_h^m(s,a)}$, the empirical transition probability; |
| $\check{P}_{h,s,a}^m$ | $\{\check{P}_{h,s,a}^m\}_{h,s,a} = \texttt{clip}\left(\left\{\left[\frac{\check{N}_h^m(s,a,s')}{\check{N}_h^m(s,a)}\right]_{s'\in\mathcal{S}}\right\}_{h,s,a}, \mathcal{W}^*\right)$; |
| $\bar{P}$ | $\bar{P} = \texttt{clip}\left(P, W^*\right)$, the clipped true transition model; |
| $\mathcal{P}^m$ | the confidence region after the $m$-th batch; |
| $\{v_h^m(s)\}$ | the extended optimal value function after the $m$-th batch; |
| $V^*\left(\bar{V}^*\right)$ | the optimal value function for the (clipped) true transition model; |
| $\alpha(n, n')$ | $\alpha(n, n') = \sqrt{\frac{4n'\iota}{n^2} + \frac{5\iota}{n}}$; |
| $\alpha^*(n, p, v)$ | $\alpha^*(n, p, v) = 5\sqrt{\frac{\mathbb{V}(p,v)\iota}{n}} + \frac{3\iota}{n}$; |

**The good event**  For $1 \leqslant m \leqslant 2H + M$, define $\mathcal{G}_{h,s,a,s'}^m$ be the event where it holds

$$\left|\hat{P}_{h,s,a,s'}^m - P_{h,s,a,s'}\right| \leqslant \beta_{h,s,a,s'}^m := \min\left\{\sqrt{\frac{2P_{h,s,a,s'}\iota}{N_h^m(s,a)}} + \frac{\iota}{3\cdot N_h^m(s,a)}, \sqrt{\frac{4\check{P}_{h,s,a,s'}^m\iota}{N_h^m(s,a)}} + \frac{5\iota}{N_h^m(s,a)}\right\}.$$

$$(24)$$

By Lemma 3 and Bernstein inequality, we have that $\mathbb{P}[\mathcal{G}_{h,s,a,s'}^m] \geqslant 1 - 2\delta$.

For $1 \leqslant m \leqslant 2H$, we set $\check{\mathcal{G}}_{h,s,a}^m$ to be the whole event. For $2H + 1 \leqslant m \leqslant M$, we define $\check{\mathcal{G}}_{h,s,a}^m$ be the event where it holds

$$(25)$$

$$\left|(\check{P}_{h,s,a} - P)v^{m-1}\right| \leqslant \lambda_{h,s,a}^m := \min\left\{5\sqrt{\frac{\mathbb{V}(\check{P}_{h,s,a}^m, v^{m-1})\iota}{\check{N}_h^m(s,a)}}\right\} \tag{26}$$

$$\left|(\check{P}_{h,s,a} - P)\bar{V}^*\right| \leqslant \lambda_{h,s,a}^{m,*} := \min\left\{5\sqrt{\frac{\mathbb{V}(\check{P}_{h,s,a}^m, \bar{V}^*)\iota}{\check{N}_h^m(s,a)}}\right\}. \tag{27}$$

Noting that $\check{P}_{h,s,a}$ is independent with both $\bar{V}^*$ and $v^{m-1}$, by Bernstein's inequality, we have that $\mathbb{P}[\check{\mathcal{G}}_{h,s,a,s'}^m] \geqslant 1 - 4\delta$

The good event $\mathcal{G}$ is defined as $\mathcal{G} = \bigcap_{h,s,a,s'} \bigcap_{m=1}^M \left( \mathcal{G}_{h,s,a,s'}^m \cap \check{\mathcal{G}}_{h,s,a}^m \right)$ Then $\mathbb{P}[\mathcal{G}] \geqslant 1 - 6S^2AHM\delta$. Throughout the analysis, we always assume $\mathcal{G}$ holds.

**Lemma 11.** *Conditioned on $\mathcal{G}$, we have $\bar{P} \in \mathcal{P}^m$ for $2H \leqslant m \leqslant 2H + M$.*

Noting that the batch complexity is bounded by $2H + M = O(H + \log_2 \log_2(K))$, it suffices to prove the regret bound. We start with counting the regret in the first two stages. The regret in the first batch is bounded by $O(H^2 k_1)$ trivially. As for the second batch, we have that

**Lemma 12.** *Conditioned on $\mathcal{G}$, with probability $1 - 4SAH\delta$ the regret bound in the second batch is bounded by $O\left( \frac{k_2\sqrt{S^4A^3H^8\iota}}{\sqrt{k_1}} + \frac{k_2S^3A^3H^4\iota}{k_1} \right)$.*

To count the regret in the third stage, we first show that the difference between the clipped model and the original model could be ignored.

**Lemma 13.** *Conditioned on $\mathcal{G}$, with probability $1 - 4S^2AH^2\delta$, for any optimal policy $\pi^*$, it holds that $\Pr_{\pi*}[\exists h \in [H], (h, s_h, a_h, s_{h+1}) \notin \mathcal{W}^*] \leqslant O\left( \frac{S^3A^2H^3\iota}{k_2} \right)$*

Based on Lemma 13, we further have that

**Lemma 14.** *Recall that $\bar{V}^*$ be the optimal value function with respect to the transition model $\bar{P}$ and reward function $r$. It then holds that $\bar{V}_1^*(s_1) \leqslant V_1^*(s_1) \leqslant \bar{V}_1^*(s_1) + O\left( \frac{S^3A^2H^4\iota}{k_2} \right)$.*

*Proof.* The left side is obvious since the reward at $z$ is always 0. On the other hand, letting $\pi^*$ be an optimal policy and $E$ be the event where $\exists h \in [H], (h, s_h, a_h, s_{h+1}) \notin \mathcal{W}^*$. Then we have that

$$
\begin{aligned}
V_1^{\pi^*}(s_1) &\leqslant \mathbb{E}_{\pi*}\left[ \left( \sum_{h=1}^H r_h(s_h, a_h) \right) \mathbb{I}[E] \right] + H\Pr_{\pi*}[E] \\
&\leqslant \mathbb{E}_{\pi*}\left[ \sum_{h=1}^H r_h(s_h, a_h)\mathbb{I}[\forall h' < h, (h', s_{h'}, a_{h'}, s_{h'+1}) \in \mathcal{W}^*] \right] + O\left( \frac{S^3A^2H^4\iota}{k_2} \right) \\
&= \bar{V}_1^{\pi^*}(s_1) + O\left( \frac{S^3A^2H^4\iota}{k_2} \right).
\end{aligned}
$$

$\square$

Recall that $\text{gap}^{m+1} := \max_{\pi \in \Pi(r, \mathcal{P}^m)}(U^\pi(\mathcal{P}^m) - L^\pi(\mathcal{P}^m))$. For $m \geqslant 2H + 1$, we have that

**Lemma 15.** *Conditioned on $\mathcal{G}$, with probability $1 - 4SAHKM\delta$, it holds that*

$$
\text{gap}^{m+1}
$$
$$
\leqslant O\left( \sqrt{\frac{SAH^3\ln(K)\iota^2}{K_{m-2H}}} + \frac{SAH^2\ln(K)\iota}{K_{m-2H}} + \sqrt{\frac{S^{\frac{11}{2}}A^4H^7\ln(K)\iota^{\frac{5}{2}}}{K_{m-2H}k_1}} + \sqrt{\frac{S^4A^{\frac{5}{2}}H^4\ln(K)\iota^{\frac{3}{2}}}{K_{m-2H}\sqrt{k_1}}} \right).
$$
(28)

By Lemma 11, 14 and 15, for any $2H \leqslant m \leqslant 2H + K$ and any $\pi \in \Pi(\mathcal{P}^m)$, we have that

$$
V_1^\pi(s_1) \geqslant L^\pi(\mathcal{P}^m) \geqslant U^\pi(\mathcal{P}^m) - \text{gap}^{m+1} \geqslant \bar{V}_1^*(s_1) - \text{gap}^{m+1} - O\left( \frac{S^3A^2H^4\iota}{k_2} \right).
$$

Recall that $k_1 = 144\sqrt{SAK\iota/H}$, $k_2 = 288S^3A^2H^4\sqrt{K\iota}$ and $K_m = \left\lceil K^{1-\frac{1}{2^m}} \right\rceil$ for $1 \leqslant m \leqslant M$. It then holds that $\frac{K_{m-2H+1}}{\sqrt{K_{m-2H}}} = \sqrt{K}$ for any $2H + 1 \leqslant m \leqslant 2H + K$. Noting that the regret in the

$m + 1$-th batch is bounded by $K_{m+1-2H} \cdot \text{gap}^{m+1}$, and the regret in the $2H + 1$-th batch is bounded by $K_1 = O(\sqrt{K})$, the total regret is bounded by

$$\text{Regret}(K) = M \cdot O\left(\sqrt{SAH^3 K \ln(K)\iota^2} + S^{\frac{15}{4}} A^{\frac{9}{8}} H^{\frac{17}{8}} \iota^{\frac{5}{8}} K^{\frac{3}{8}} + S^{\frac{19}{4}} A^{\frac{13}{4}} H^{\frac{33}{4}} \ln(K)\iota K^{\frac{1}{4}} + S^{\frac{11}{2}} A^{\frac{9}{2}} H^{\frac{17}{2}} \iota\right).$$

By replacing $\delta$ by $\frac{\delta}{20S^2 AHK}$, we get the desired regret bound.

Below we analyze the computational cost of Algorithm 1. By Lemma 2 the computational costs of Sum is $O(nS^3 A^2 H^2)$, where $n$ is the number of inputs for Sum.

Below we analyze the computational cost of PolicySearch. By Lemma 8, for input $(u, u', \mathcal{P})$, the computational cost of PolicySearch is bounded by $O(S^4 AHM^3 \log(SAHK) \log(SAHK/(a - b)))$ with $a = \max_\pi U^\pi(u + \mathbf{1}_z, \mathcal{P})$ and $b = \max_\pi L^\pi(u, \mathcal{P})$.

In the first stage, we invoke PolicySearch with $u = 0$, which implies $b = 0$ and $W^\pi(u, p) = 0$ for any $\pi$ and $p \in \mathcal{P}$. Then the condition in Line 7 Algorithm 4 is satisfied and the loop would break. Therefore, by Lemma 10, the computational cost of PolicySearch in the first stage is bounded by $O(S^4 AHM^3 \log(SAKH))$.

In the second and the third stage, we invoke PolicySearch with $u = r$. In this case, if $a - b \leqslant 1/K$, then we can learn an $1/K$-optimal policy by solving $\pi' = \arg\max_\pi L^\pi(r, \mathcal{P})$. Then we can simply run this policy in the left episodes. Without loss of generality, we then assume that $a - b > 1/K$, which implies the time cost of PolicySearch is bounded by $O(S^4 AHM^3 \log^2(SAKH))$.

Now we count the number of callings to Sum and PolicySearch. In the first and second stage, Sum is called for $2H$ times with $n = SAH$ inputs, and PolicySearch is called for $2H$ times. In the third stage, Sum is called for $M$ times with $n = K^3$ inputs, and PolicySearch is called for $K^3 M$ times. So the total time cost due to Sum and PolicySearch is bounded by $\tilde{O}(S^4 AHK^3 + S^3 A^2 H^2 K^3)$. On the other hand, to compute $\{v_h^m(s)\}_{h\in[H], s\in\mathcal{S}}$ in Line 15 Algorithm 4, we need to invoke EVI (see Algorithm 5) for $M$ times, which needs additional $O(S^4 AHM^4 \log(SAHK))$ time by Lemma 10. Finally, to observe the samples and compute the confidence region, we need $O(S^2 AHK)$ time.

Putting all together, the computational cost of Algorithm 1 is bounded by $\tilde{O}(S^4 AHK^3 + S^3 A^2 H^2 K^3)$. The proof is completed.

### D.1 Proof of Lemma 11

**Lemma 11 (restated)** *Conditioned on $\mathcal{G}$, we have $\bar{P} \in \mathcal{P}^m$ for $2H \leqslant m \leqslant 2H + M$.*

*Proof.* with a slight abuse of notation, we use $v^m$ to denote the value of $v^{m-2H}$ in Algorithm 3.

Recall the definition of $\mathcal{P}^m$. It suffices to show that $\bar{P} \in \text{CR}^*(\mathcal{D}^m, \mathcal{D}^m, W^*, \{\bar{v}_h^{m-1}(s)\}_{(h,s)})$ for each $m \geqslant 2H$.

Note that after the $m$-th batch $\hat{p}_{h,s,a,s'} = \hat{P}_{h,s,a,s'}^m$ and $\check{p}_{h,s,a} = \check{P}_{h,s,a}^m$. By the definition of $\mathcal{G}$, and recalling the definition of $\beta_{h,s,a,s'}^m$ and $\lambda_{h,s,a}^m$ in (24) and (26), we have that

$$\left|\bar{P}_{h,s,a,s'} - \hat{P}_{h,s,a,s'}^m\right| \leqslant \beta_{h,s,a,s'}^m \leqslant \alpha(N_h^m(s,a), N_h^m(s,a,s'))$$
$$\left|(\bar{P}_{h,s,a} - \check{P}_{h,s,a}^m)v^{m-1}\right| \leqslant \lambda_{h,s,a}^m \leqslant \alpha^*(\check{N}_h^m(s,a), \check{P}_{h,s,a}^m, v^{m-1}).$$

The proof is completed. $\square$

### D.2 Proof of Lemma 12

**Lemma 12 (restated)** *Conditioned on $\mathcal{G}$, with probability $1 - 4SAH\delta$ the regret bound in the second stage is bounded by $O\left(\frac{k_2 \sqrt{S^4 A^3 H^8 \iota}}{\sqrt{k_1}} + \frac{k_2 S^3 A^3 H^4 \iota}{k_1}\right)$.*

*Proof.* Let $\mathcal{D}^1$ and $\mathcal{D}^2$ be respectively the dataset after the first and second stage. Let $\{\bar{N}_h^1(s,a,s')\}$ and $\{\bar{N}_h^2(s,a,s')\}$ be the corresponding counts. Let $\bar{\mathcal{W}}^1$ and $\bar{\mathcal{W}}^2$ be the corresponding *known* set.

Note that $\mathcal{W}^* = \bar{\mathcal{W}}^2$. By Lemma 16, with probability $1 - 8S^2AH^2\delta$, it holds that

$$\max_\pi \mathbb{P}_\pi \left[ \exists h \in [H], (h, s_h, a_h, s_{h+1}) \notin \bar{\mathcal{W}}^1 \right] \leqslant \frac{36C_1 S^2 A^2 H^3 \iota}{k_1}$$

$$\bar{N}_h^1(s, a) \geqslant \frac{ck}{27SA} \max_\pi W^\pi(\mathbf{1}_{h,s,a}, P) - 4\iota - \frac{36C_1 SAH^3 \iota}{27}. \tag{29}$$

For any policy $\pi$ in $\Pi(\mathrm{CR}(\mathcal{D}^1))$, using policy difference lemma we have that

$U^\pi(\mathrm{CR}(\mathcal{D}^1)) - L^\pi(\mathrm{CR}(\mathcal{D}^1))$

$$= U^\pi(\mathrm{CR}(\mathcal{D}^1)) - W^\pi(r, \mathtt{clip}(P, \bar{\mathcal{W}}^1)) + W^\pi(r, \mathtt{clip}(P, \bar{\mathcal{W}}^1)) - L^\pi(\mathrm{CR}(\mathcal{D}^1)) \tag{30}$$

$$\leqslant \max_\pi \mathbb{P}_\pi \left[ \exists h \in [H], (h, s_h, a_h, s_{h+1}) \notin \bar{\mathcal{W}}^1 \right] + O\left( \sum_{h,s,a} W^\pi(\mathbf{1}_{h,s,a}, \mathtt{clip}(P, \bar{\mathcal{W}}^1)) \sqrt{\frac{S\iota}{\bar{N}_h^1(s,a)}} \cdot H \right)$$

$$\leqslant \frac{36C_1 S^2 A^2 H^3 \iota}{k_1} + O\left( \sum_{h,s,a} \left( \frac{SA(\bar{N}_h^1(s,a) + SAH^3\iota)}{k} \right) \sqrt{\frac{SH^2\iota}{\bar{N}_h^1(s,a)}} \right) \tag{31}$$

$$\leqslant \frac{36C_1 S^2 A^2 H^3 \iota}{k_1} + O\left( \sqrt{\frac{S^4 A^3 H^8 \iota}{k_1}} + \frac{S^3 A^3 H^4 \iota}{k_1} \right),$$

where the third line is by (29) and the last line is by Cauchy's inequality and the fact that $\bar{N}_h^1(s,a) \geqslant 1$. Conditioned on $\mathcal{G}$, we have that $= \mathtt{clip}(P, \bar{\mathcal{W}}^1) \in \mathrm{CR}(\mathcal{D}^1)$. As a result, we have that $\max_\pi U^\pi(\mathrm{CR}(\mathcal{D}^1)) \geqslant V_1^*(s_1) - \frac{36C_1 S^3 A^2 H^4 \iota}{k_1}$. To conclude, the regret in the second stage is bounded by $O\left( \frac{k_2 \sqrt{S^4 A^3 H^8 \iota}}{\sqrt{k_1}} + \frac{k_2 S^3 A^3 H^4 \iota}{k_1} \right)$. $\qquad\square$

### D.3 Proof of Lemma 13

**Lemma 13 (restated)** *Conditioned on $\mathcal{G}$, with probability $1 - 4S^2AH^2\delta$, for any optimal policy $\pi^*$, it holds that $\mathrm{Pr}_{\pi^*}[\exists h \in [H], (h, s_h, a_h, s_{h+1}) \notin \mathcal{W}^*] \leqslant O\left( \frac{S^3 A^2 H^3 \iota}{k_2} \right)$.*

*Proof.* By Lemma 16, with probability $1 - 4S^2AH^2\delta$, it holds that

$$\max_{\pi \in \Pi^*} \mathrm{Pr}_\pi \left[ \exists h \in [H], (h, s_h, a_h, s_{h+1}) \notin \mathcal{W}^* \right] \leqslant \frac{36C_1 S^2 A^2 H^3 \iota}{k_2}.$$

The proof is completed. $\qquad\square$

### D.4 Proof of Lemma 15

**Lemma 15 (restated)** *Conditioned on $\mathcal{G}$, with probability $1 - 4SAHKM\delta$, it holds that*

$$\mathrm{gap}^m \leqslant O\left( \sqrt{\frac{SAH^3 \ln(K)\iota^2}{K_{m-2H}}} + \sqrt{\frac{SAH^2 \ln(K)\iota}{K_{m-2H}}} + \sqrt{\frac{S^{\frac{11}{2}} A^4 H^7 \ln(K)\iota^{\frac{5}{2}}}{K_{m-2H} k_1}} + \sqrt{\frac{S^4 A^{\frac{5}{2}} H^4 \ln(K)\iota^{\frac{3}{2}}}{K_{m-2H} \sqrt{k_1}}} \right)$$

*for $2H + 1 \leqslant m \leqslant M$.*

*Proof.* Let $m \in [2H + 1, M]$ be fixed. Conditioned on $\mathcal{G}$, we have that for any $p \in \mathcal{P}^{m-1}$, for any $(h, s, a, s') \in \mathcal{W}^*$ it holds that

$$\left| \hat{P}_{h,s,a,s'}^{m-1} - \bar{P}_{h,s,a,s'} \right| \leqslant \sqrt{\frac{4\hat{P}_{h,s,a,s'}^{m-1}\iota}{N_h^{m-1}(s,a)}} + \frac{\iota}{3N_h^{m-1}(s,a)}$$

$$= \frac{1}{N_h^{m-1}(s,a)} \cdot \left( \sqrt{4N_h^{m-1}(s,a,s')\iota} + 1/3 \right)$$

$$\leqslant 3\hat{P}_{h,s,a,s'}^{m-1} \cdot \sqrt{\frac{\iota}{N_h^{m-1}(s,a,s')}}$$

$$\leqslant \frac{1}{3H} \hat{P}_{h,s,a,s'}^{m-1}.$$

On the other hand, noting that for any $p \in \mathcal{P}^{m-1}$ and $(h, s, a, s') \in \mathcal{W}^*$, with similar computation it holds that

$$\left| p_{h,s,a,s'} - \bar{P}_{h,s,a,s'} \right| \leqslant \left| p_{h,s,a,s'} - \hat{P}_{h,s,a,s'}^{m-1} \right| + \left| \hat{p}_{h,s,a,s'} - \bar{P}_{h,s,a,s'}^{h'} \right|$$

$$\leqslant \frac{1}{3H} \bar{P}_{h,s,a,s'} + \frac{1}{3H} \hat{P}_{h,s,a,s'}^{m-1}$$

$$\leqslant \left( \frac{2}{3H} + \frac{1}{9H^2} \right) \bar{P}_{h,s,a,s'}$$

Therefore $\mathcal{P}^{m-1}$ is *tight* with respect to $\bar{P}$. Let $p^{m-1} \in \mathcal{P}^{m-1}$ be the value of $p$ in Line 26 Algorithm 3. Let $r^{i,m-1}$ be the value of $r^i$ defined in Line 29 Algorithm 3. Let $\{\tilde{\pi}^{i,m-1}\}$ be the value of $\tilde{\pi}(i)$ in Line 30 Algorithm 3.

As a result, by Lemma 8, Lemma 2 and Lemma 17

$$W^{\tilde{\pi}^{i,m-1}}(r^{i,m-1}, \bar{P}) \geqslant \frac{c}{9} \max_{\pi \in \Pi(\mathcal{P}^{m-1})} W^\pi(r^{i,m-1}, \bar{P})$$

$$W^{\pi^m}(\mathbf{1}_{h,s,a}, \bar{P}) \geqslant \frac{1}{9K^3} \sum_{i=1}^{K^3} W^{\tilde{\pi}^{i,m-1}}(\mathbf{1}_{h,s,a}, \bar{P}), \forall (h, s, a). \tag{32}$$

Consequently, for any $\pi \in \Pi(\mathcal{P}^{m-1})$ and $(h, s, a)$, it holds that

$$W^\pi(r^{K^3+1,m-1}, \bar{P}) \leqslant \frac{81}{cK^3} \sum_{i=1}^{K^3} W^{\tilde{\pi}^{i,m-1}}(r^{i,m-1}, \bar{P})$$

$$= \frac{81}{cK^3} \sum_{i=1}^{K^3} \sum_{h,s,a} W^{\tilde{\pi}^{i,m-1}}(\mathbf{1}_{h,s,a}, \bar{P}) \cdot \min \left\{ \frac{1}{\sum_{j=1}^{i-1} W^{\tilde{\pi}^{j,m-1}}(\mathbf{1}_{h,s,a}, p^{m-1})}, 1 \right\}$$

$$= \frac{81}{cK^3} \sum_{h,s,a} \sum_{i=1}^{K^3} \sum_{h,s,a} W^{\tilde{\pi}^{i,m-1}}(\mathbf{1}_{h,s,a}, \bar{P}) \cdot \min \left\{ \frac{1}{\sum_{j=1}^{i-1} W^{\tilde{\pi}^{j,m-1}}(\mathbf{1}_{h,s,a}, p^{m-1})}, 1 \right\}$$

$$\leqslant \frac{243}{cK^3} \sum_{h,s,a} \sum_{i=1}^{K^3} \sum_{h,s,a} W^{\tilde{\pi}^{i,m-1}}(\mathbf{1}_{h,s,a}, \bar{P}) \cdot \min \left\{ \frac{1}{\sum_{j=1}^{i-1} W^{\tilde{\pi}^{j,m-1}}(\mathbf{1}_{h,s,a}, \bar{P})}, 1 \right\}$$

$$\leqslant \frac{243SAH \ln(K)}{cK^3} \tag{33}$$

where the second line is by the *tightness* (w.r.t. $\bar{P}$) of $\mathcal{P}^{m-1}$, and the last line is by the fact that for any non-negative $\{x_i\}_{i=1}^n$

$$\sum_{i=1}^n x_i \cdot \min \left\{ \frac{1}{\sum_{j=1}^{i-1} x_j}, 1 \right\} \leqslant 2 + 2 \sum_{i=1}^n \left( \ln \left( \sum_{j=1}^i x_j \right) - \ln \left( \sum_{j=1}^{i-1} x_j \right) \right) \mathbb{I} \left[ \left( \sum_{j=1}^{i-1} x_j \right) \geqslant 1 \right]$$

$$\leqslant 2 + 2 \ln \left( \sum_{i=1}^n x_i \right).$$

By definition of $r^{K^3,m-1}$, we have that for any $(h, s, a)$

$$r_h^{K^3+1,m-1}(s, a) = \min \left\{ \frac{1}{\sum_{j=1}^{K^3} W^{\tilde{\pi}^{j,m-1}}(\mathbf{1}_{h,s,a}, p^{m-1})}, 1 \right\}$$

$$\geqslant \frac{1}{3} \min \left\{ \frac{1}{\sum_{j=1}^{K^3} W^{\tilde{\pi}^{j,m-1}}(\mathbf{1}_{h,s,a}, \bar{P})}, 1 \right\} = \frac{1}{3} \min \left\{ \frac{1}{K^3 W^{\pi^m}(\mathbf{1}_{h,s,a}, \bar{P})}, 1 \right\}. \tag{34}$$

By (33) and (34), for any $\pi \in \Pi(\mathcal{P}^{m-1})$ it holds that

$$\sum_{h,s,a} W^\pi(\mathbf{1}_{h,s,a}, \bar{P}) \cdot \min \left\{ \frac{1}{K^3 W^{\pi^m}(\mathbf{1}_{h,s,a}, \bar{P})}, 1 \right\} \leqslant 3 \sum_{h,s,a} W^\pi(\mathbf{1}_{h,s,a}, \bar{P}) r_h^{K^3+1,m-1}(s, a) \leqslant \frac{729SAH \ln(K)}{cK^3}. \tag{35}$$

Note that $\pi^m$ is executed for $K_{m-2H}$ rounds. By Lemma 4, with probability $1 - 4SAH\delta$, it holds that

$$\check{N}_h^m(s,a) \geqslant \frac{1}{3}K_{m-2H}W^{\pi^m}(\mathbf{1}_{h,s,a},\bar{P}) - \iota. \tag{36}$$

Fix $\pi \in \Pi(r,\mathcal{P}^{m-1})$. Let $\{f_h(\cdot)\}_{h=1}^S$ be the value function under $\pi$ and $\bar{P}$. For any $P' \in \mathcal{P}^m$, by policy difference lemma, we have that

$$\left|W^\pi(r,P') - W^\pi(r,\bar{P})\right|$$

$$= \left|\sum_{h,s,a} W^\pi(\mathbf{1}_{h,s,a},P') \cdot (P'_{h,s,a} - \bar{P}_{h,s,a})f_{h+1}\right|$$

$$\leqslant \underbrace{\left|\sum_{h,s,a} W^\pi(\mathbf{1}_{h,s,a},P')(P'_{h,s,a} - \bar{P}_{h,s,a})v_{h+1}^{m-1}\right|}_{\textbf{Term.1}} + \underbrace{\left|\sum_{h,s,a} W^\pi(\mathbf{1}_{h,s,a},P')(P'_{h,s,a} - \bar{P}_{h,s,a})(f_{h+1} - v_{h+1}^{m-1})\right|}_{\textbf{Term.2}}. \tag{37}$$

By the definition of $\mathcal{P}^m$ and $\mathcal{G}$, we have that

$$\textbf{Term.1} = \left|\sum_{h,s,a} W^\pi(\mathbf{1}_{h,s,a},P')(P'_{h,s,a} - \check{P}_{h,s,a}^m + \check{P}_{h,s,a}^m - P_{h,s,a})v_{h+1}^{m-1}\right|$$

$$\leqslant \sum_{h,s,a} W^\pi(\mathbf{1}_{h,s,a},P') \cdot \left(5\sqrt{\frac{\mathbb{V}(\check{P}_{h,s,a}^m,v_{h+1}^{m-1})\iota}{\check{N}_h^m(s,a)}} + 5\sqrt{\frac{\mathbb{V}(\bar{P}_{h,s,a},v_{h+1}^{m-1})\iota}{\check{N}_h^m(s,a)}} + \frac{8\iota}{\check{N}_h^m(s,a)}\right)$$

$$\leqslant O\left(\sqrt{\sum_{h,s,a}\frac{W^\pi(\mathbf{1}_{h,s,a},\bar{P})\iota}{\check{N}_h^m(s,a)}} \cdot \sqrt{\sum_{h,s,a}W^\pi(\mathbf{1}_{h,s,a},\bar{P})\cdot\left(\mathbb{V}(\check{P}_{h,s,a}^m,v_{h+1}^{m-1}) + \mathbb{V}(\bar{P}_{h,s,a},v_{h+1}^{m-1})\right)}\right)$$

$$+ O\left(\sum_{h,s,a}\frac{W^\pi(\mathbf{1}_{h,s,a},\bar{P})\iota}{\check{N}_h^m(s,a)}\right) \tag{38}$$

Define $T_1 = \sum_{h,s,a}\frac{W^\pi(\mathbf{1}_{h,s,a},\bar{P})\iota}{\check{N}_h^m(s,a)}$, $T_2 = \sum_{h,s,a}W^\pi(\mathbf{1}_{h,s,a},\bar{P})\cdot\mathbb{V}(\bar{P}_{h,s,a},v_{h+1}^{m-1})$ and $T_2 = \sum_{h,s,a}W^\pi(\mathbf{1}_{h,s,a},\bar{P})\cdot\mathbb{V}(\check{P}_{h,s,a}^m,v_{h+1}^{m-1})$.

**Bound of $T_1$** By (35) and (36), we have that

$$T_1 \leqslant 3\sum_{h,s,a}\frac{W^\pi(\mathbf{1}_{h,s,a},\bar{P})}{\max\{K_{m-2H}W^{\pi^m}(\mathbf{1}_{h,s,a},\bar{P}) - 3\iota, 1\}}$$

$$= \frac{3K^3}{K_{m-2H}}\sum_{h,s,a}W^\pi(\mathbf{1}_{h,s,a},\bar{P})\cdot\min\left\{\frac{1}{K^3W^{\pi^m}(\mathbf{1}_{h,s,a},\bar{P}) - 3K^3\iota/K_{m-2H}}, \frac{K_{m-2H}}{K^3}\right\}$$

$$\leqslant \frac{3K^3}{K_{m-2H}}\sum_{h,s,a}W^\pi(\mathbf{1}_{h,s,a},\bar{P})\cdot\left(\min\left\{\frac{2}{K^3W^{\pi^m}(\mathbf{1}_{h,s,a},\bar{P})}, 1\right\}\cdot\mathbb{I}\left[K_{m-2H}W^{\pi^m}(\mathbf{1}_{h,s,a},\bar{P}) \geqslant 6\iota\right]\right)$$

$$+ 3\sum_{h,s,a}W^\pi(\mathbf{1}_{h,s,a},\bar{P})\mathbb{I}\left[K_{m-2H}W^{\pi^m}(\mathbf{1}_{h,s,a},\bar{P}) < 6\iota\right]$$

$$\leqslant \frac{3K^3}{K_{m-2H}}\cdot\frac{729SAH\ln(K)}{cK^3} + \frac{18SAH\iota}{K_{m-2H}}$$

$$= O\left(\frac{SAH\ln(K)\iota}{K_{m-2H}}\right). \tag{39}$$

**Bound of $T_2$**

$$T_2 = \sum_{h,s,a} W^\pi(\mathbf{1}_{h,s,a}, \bar{P}) \cdot \mathbb{V}(\bar{P}_{h,s,a}, v_{h+1}^{m-1})$$

$$= \sum_{h,s,a} W^\pi(\mathbf{1}_{h,s,a}, \bar{P}) \cdot \left(\bar{P}_{h,s,a}(v_{h+1}^{m-1})^2 - (\bar{P}_{h,s,a}v_{h+1}^{m-1})^2\right)$$

$$\leqslant \sum_{h,s,a} W^\pi(\mathbf{1}_{h,s,a}, \bar{P}) \cdot \left((v_h^{m-1}(s))^2 - (\bar{P}_{h,s,a}v_{h+1}^{m-1})^2\right) + H^2$$

$$\leqslant H \sum_{h=1}^{H} \mathbb{E}_{\pi,\bar{P}} \left[|v_h^{m-1}(s_h) - \bar{P}_{h,s_h,a_h}v_{h+1}^{m-1}|\right] + H^2$$

$$= H \sum_{h=1}^{H} \mathbb{E}_{\pi,\bar{P}} \left[v_h^{m-1}(s_h) - \bar{P}_{h,s_h,a_h}v_{h+1}^{m-1}\right] + H^2 \tag{40}$$

$$\leqslant H \sum_{h=1}^{H} \mathbb{E}_{\pi,\bar{P}}[r_h(s_h, a_h)] + 2H^2$$

$$\leqslant 4H^2. \tag{41}$$

Here (40) is by the fact that $v_h^{m-1}$ is the optimal value function with respect to $\mathcal{P}^{m-1}$ and $\bar{P} \in \mathcal{P}^{m-1}$.

**Bound of $T_3$**  By Lemma 3, with probability $1 - 4S^2AH\delta$, it holds that

$$\left|\check{P}_{h,s,a,s'}^m - \bar{P}_{h,s,a,s'}\right| \leqslant 4\sqrt{\frac{\bar{P}_{h,s,a,s'}\iota}{\check{N}_h^m(s,a)}} + \frac{3\iota}{\check{N}_h^m(s,a)} \leqslant 2\bar{P}_{h,s,a,s'} + \frac{5\iota}{\check{N}_h^m(s,a)}.$$

As a result, we have that

$$T_3 = \sum_{h,s,a} W^\pi(\mathbf{1}_{h,s,a}, \bar{P}) \cdot \mathbb{V}(\check{P}_{h,s,a}^m, v_{h+1}^{m-1})$$

$$\leqslant \sum_{h,s,a} W^\pi(\mathbf{1}_{h,s,a}, \bar{P}) \cdot \sum_{s'} \check{P}_{h,s,a,s'}^m \left(v_{h+1}^{m-1}(s') - \bar{P}_{h,s,a,s'}v_{h+1}^{m-1}\right)^2$$

$$\leqslant \sum_{h,s,a} W^\pi(\mathbf{1}_{h,s,a}, \bar{P}) \cdot \sum_{s'} \bar{P}_{h,s,a,s'}^m \left(v_{h+1}^{m-1}(s') - \bar{P}_{h,s,a,s'}v_{h+1}^{m-1}\right)^2 + \sum_{h,s,a} W^\pi(\mathbf{1}_{h,s,a}, \bar{P}) \cdot \frac{5H^2\iota}{\check{N}_h^m(s,a)}$$

$$= 4T_2 + 5H^2\iota T_1$$

$$\leqslant O\left(H^2 + \frac{SAH^2\ln(K)\iota^2}{K_{2m-H}}\right). \tag{42}$$

By (39), (41) and (42), **Term.1** is bounded by

$$\mathbf{Term.1} \leqslant O\left(\sqrt{\frac{SAH^3\ln(K)\iota^2}{K_{m-2H}}} + \frac{SAH^2\ln(K)\iota}{K_{m-2H}}\right). \tag{43}$$

To bound **Term.2**, by definition of $\mathcal{P}^m$ and $\mathcal{G}$, we have

$$\mathbf{Term.2} = \left| \sum_{h,s,a} W^\pi(\mathbf{1}_{h,s,a}, P')(P'_{h,s,a} - \bar{P}_{h,s,a})(f_{h+1} - v_{h+1}^{m-1}) \right|$$

$$\leqslant \sum_{h,s,a} W^\pi(\mathbf{1}_{h,s,a}, P') \sum_{s'} \left( 10\sqrt{\frac{\bar{P}_{h,s,a,s'}\iota}{N_h^m(s,a)}} + \frac{6\iota}{N_h^m(s,a)} \right) \cdot |f_{h+1}(s') - v_{h+1}^{m-1}(s') - l|$$

$$\leqslant O\left( \sum_{h,s,a} W^\pi(\mathbf{1}_{h,s,a}, \bar{P}) \sum_{s'} \sqrt{\frac{\bar{P}_{h,s,a,s'}\iota}{N_h^m(s,a)}} |f_{h+1}(s') - v_{h+1}^{m-1}(s') - l_h(s,a)| \right)$$

$$+ O\left( \sum_{h,s,a} W^\pi(\mathbf{1}_{h,s,a}, \bar{P}) \frac{SH\iota}{N_h^m(s,a)} \right),$$

(44)

where $l_h(s,a) = \bar{P}_{h,s,a}(f_{h+1} - v_{h+1}^m)$. By (39), the second term in (44) is bounded by $O\left( \frac{SAH\ln(K)\iota}{K_{m-2H}} \right)$. To bound the the first term in (44), by Cauchy's inequality, we have that

$$O\left( \sum_{h,s,a} W^\pi(\mathbf{1}_{h,s,a}, \bar{P})\sqrt{\frac{S\mathbb{V}(\bar{P}_{h,s,a}, f_{h+1} - v_{h+1}^{m-1})\iota}{N_h^m(s,a)}} \right)$$

$$\leqslant O\left( \sqrt{\frac{SW^\pi(\mathbf{1}_{h,s,a}, \bar{P})\iota}{N_h^m(s,a)}} \cdot \sqrt{\sum_{h,s,a} W^\pi(\mathbf{1}_{h,s,a}, \bar{P})\mathbb{V}(\bar{P}_{h,s,a}, f_{h+1} - v_{h+1}^{m-1})} \right)$$

$$\leqslant O\left( \sqrt{\frac{S^2 AH\ln(K)\iota^2}{K_{m-2H}}} \cdot \sqrt{\sum_{h,s,a} W^\pi(\mathbf{1}_{h,s,a}, \bar{P})\mathbb{V}(\bar{P}_{h,s,a}, f_{h+1} - v_{h+1}^{m-1})} \right),$$

where the last line is by (39). Continuing the computation:

$$\sum_{h,s,a} W^\pi(\mathbf{1}_{h,s,a}, \bar{P})\mathbb{V}(\bar{P}_{h,s,a}, f_{h+1} - v_{h+1}^{m-1})$$

$$= \sum_{h,s,a} W^\pi(\mathbf{1}_{h,s,a}, \bar{P})\left( \bar{P}_{h,s,a}(f_{h+1} - v_{h+1^{m-1}})^2 - (\bar{P}_{h,s,a}f_{h+1} - \bar{P}_{h,s,a}v_{h+1}^{m-1})^2 \right)$$

$$\leqslant \mathbb{E}_{\pi,\bar{P}}\left[ \sum_{h=1}^H \left( (f_{h+1}(s_{h+1}) - v_{h+1}^{m-1}(s_{h+1})^2 - (\sum_a \pi_h(a|s)\bar{P}_{h,s,a}(f_{h+1} - v_{h+1}))^2 \right) \right] \quad (45)$$

$$\leqslant (v_1^{m-1}(s_1) - f_1(s_1))^2 + H\mathbb{E}_{\pi,\bar{P}}\left[ \sum_{h=1}^H \left| f_h(s_h) - v_h^{m-1}(s_h) - \sum_a \pi_h(a|s_h)\bar{P}_{h,s_h,a}(f_{h+1} - v_{h+1}^{m-1}) \right| \right]$$

$$\leqslant (v_1^{m-1}(s_1) - f_1(s_1))^2 + H\mathbb{E}_{\pi,\bar{P}}\left[ \sum_{h=1}^H v_h^{m-1}(s_h) - \sum_a \pi_h(a|s_h)\left( r_h(s_h,a) + \bar{P}_{h,s_h,a}v_{h+1}^{m-1} \right) \right]$$

(46)

$$\leqslant (v_1^{m-1}(s_1) - f_1(s_1))^2 + H(v_1^{m-1}(s_1) - f_1(s_1))$$

$$\leqslant 2H(v_1^{m-1}(s_1) - f_1(s_1)).$$

Here (45) holds by the fact that $\text{Var}(X) \geqslant \mathbb{E}_Y[\text{Var}(X|Y)]$ for any random variables $X$ and $Y$ (recalling that $\text{Var}(X)$ denotes the variance of $X$), and (46) is by the fact that $v_h^{m-1}(s_h) \geqslant \sum_a \pi_h(a|s_h)(r_h(s_h,a) + \bar{P}_{h,s,a}v_{h+1}^{m-1})$ and $f_h(s_h) = \sum_a \pi_h(a|s_h)(r_h(s_h,a) + \bar{P}_{h,s,a}f_{h+1})$ for any $1 \leqslant h \leqslant H$.

Because $\pi \in \Pi(r, \mathcal{P}^{m-1})$, we learn that $v_1^{m-1} - f_1(s_1) \leqslant \text{gap}^m$. By Lemma 16, we have that for any $m \geqslant H + 1$, $N_h^{m-1}(s,a) \geqslant \frac{ck_1}{27SA}\max_\pi W^\pi(\mathbf{1}_{h,s,a}, \bar{P}) - 4\iota - \frac{36C_1 SAH^3\iota}{27}$. With similar

analysis, and noting that $\|p'_{h,s,a} - p''_{h,s,a}\|_1 \leqslant O(\sqrt{S\iota/N_h^{m-1}(s,a)})$ for any $p', p'' \in \mathcal{P}^{m-1}$, we have

$$\text{gap}^m \leqslant O\left(\max_{\pi'} \sum_{h,s,a} W^{\pi'}(\mathbf{1}_{h,s,a}, \bar{P})\sqrt{\frac{SH^2\iota}{N_h^{m-1}(s,a)}}\right)$$

$$\leqslant O\left(\sqrt{\frac{S^4 A^3 H^4 \iota}{k_1}} + \frac{S^{\frac{7}{2}} A^3 H^5 \iota^{\frac{3}{2}}}{k_1}\right). \tag{47}$$

As a result, we have that

$$\textbf{Term.2} \leqslant O\left(\sqrt{\frac{S^{\frac{11}{2}} A^4 H^7 \ln(K)\iota^{\frac{5}{2}}}{K_{m-2H}k_1}} + \sqrt{\frac{S^4 A^{\frac{5}{2}} H^4 \ln(K)\iota^{\frac{3}{2}}}{K_{m-2H}\sqrt{k_1}}} + \frac{S^2 AH \ln(K)\iota}{K_{m-2H}}\right). \tag{48}$$

Putting all together, for any $\pi \in \Pi(r, \mathcal{P}^{m-1})$ and any $P' \in \mathcal{P}^m$, we have

$$|W^\pi(r, P') - W^\pi(r, \bar{P})|$$

$$\leqslant O\left(\sqrt{\frac{SAH^3 \ln(K)\iota^2}{K_{m-2H}}} + \frac{SAH^2 \ln(K)\iota}{K_{m-2H}} + \sqrt{\frac{S^{\frac{11}{2}} A^4 H^7 \ln(K)\iota^{\frac{5}{2}}}{K_{m-2H}k_1}} + \sqrt{\frac{S^4 A^{\frac{5}{2}} H^4 \ln(K)\iota^{\frac{3}{2}}}{K_{m-2H}\sqrt{k_1}}}\right).$$

By definition, there exists $P', P''$ such that $U^\pi(\mathcal{P}^m) = W^\pi(r, P')$ and $L^\pi(\mathcal{P}^m) = W^\pi(r, P'')$. Therefore,

$$|U^\pi(\mathcal{P}^m) - L^\pi(\mathcal{P}^m)|$$

$$\leqslant O\left(\sqrt{\frac{SAH^3 \ln(K)\iota^2}{K_{m-2H}}} + \frac{SAH^2 \ln(K)\iota}{K_{m-2H}} + \sqrt{\frac{S^{\frac{11}{2}} A^4 H^7 \ln(K)\iota^{\frac{5}{2}}}{K_{m-2H}k_1}} + \sqrt{\frac{S^4 A^{\frac{5}{2}} H^4 \ln(K)\iota^{\frac{3}{2}}}{K_{m-2H}\sqrt{k_1}}}\right).$$

Taking maximization over $\pi \in \Pi(r, \mathcal{P}^{m-1})$ we finish the proof. $\qquad\square$

### D.4.1 Statement and Proof of Lemma 16

**Lemma 16.** *Given a dataset $\mathcal{D}$ and $k \geqslant 0$, let $\mathcal{D}'$ be the output by running Algorithm 2 with input $(r, \mathcal{D}, k)$. Let $\{N_h(s, a, s')\}(\{N_h'(s, a, s')\})$ be the counts with respect to $\mathcal{D}(\mathcal{D}')$. Let $\mathcal{W} = \{(h, s, a, s')|N_h(s, a, s') \geqslant C_1 H^2\iota\}$ and $\mathcal{W}' = \{(h, s, a, s')|N_h'(s, a, s') \geqslant C_1 H^2\iota\}$. Let $\bar{p} = \texttt{clip}(P, \mathcal{W})$. With probability $1 - 4S^2 AH^2\delta$, it holds that*

$$\max_{\pi \in \Pi*} \Pr_\pi\left[\exists h' \in [h], (h', s_{h'}, a_{h'}, s_{h'+1}) \notin \mathcal{W}'\right] \leqslant \frac{36C_1 S^2 A^2 H^3\iota}{k}, \tag{49}$$

*where $\Pi*$ is the set of optimal policies. Moreover, if $\mathcal{D} = \varnothing$ and $u = 0$, with probability $1 - 4S^2 AH^2\delta$ it holds that*

$$N'_{h,s,a} \geqslant \frac{ck}{27SA} \max_\pi W^\pi(\mathbf{1}_{h,s,a}, P) - 4\iota - \frac{36C_1 SAH^3\iota}{27} \tag{50}$$

*for any $1 \leqslant h \leqslant H$.*

*Proof.* For $h' = 1, 2, ..., H$, we denote $\mathcal{D}^{h'}$ as the value of $\mathcal{D}$ after the $h'$-th batch in Algorithm 2. Similarly, we define $\{N_h^{h'}(s, a, s')\}$, $\{N_h^{h'}(s, a)\}$ and $\{\hat{p}_{h,s,a}^{h'}\}$ be respectively the value of $\{N_h(s, a, s')\}$, $\{N_h(s, a)\}$ and $\{\hat{p}_{h,s,a}\}$ after the $h'$-th batch. Note that $\mathcal{P}^{h'} = \texttt{CR}(\mathcal{D}^{h'})$ is the value of $\mathcal{P}$ after the $h'$-th batch.

Define $\mathcal{W}^{h'} := \{(h, s, a, s') : N_h^{h'}(s, a, s') \geqslant C_1 H^2\iota\}$ and $P^{h'} = \texttt{clip}(P, \mathcal{W}^{h'})$. Let $p^{h'} \in \mathcal{P}^{h'-1}$ be the transition model chosen at line 8 Algorithm 2.

Using Lemma 3 and Lemma 4, with probability $1 - 4S^2AH^2\delta$, for any $(h, s, a, s') \in \mathcal{W}^{h'}$, it holds that

$$\left| P_{h,s,a,s'}^{h'} - \hat{p}_{h,s,a,s'}^{h'} \right| \leqslant \sqrt{\frac{4P_{h,s,a,s'}^{h'}\iota}{N_h^{h'}(s,a)}} + \frac{\iota}{3N_h^{h'}(s,a)}$$

$$\leqslant \frac{1}{3H}P_{h,s,a,s'}^{h'},$$

where in the last inequality, we use Lemma 4 to get that $N_h^{h'}P_{h,s,a,s'}^{h'} \geqslant \frac{1}{3}N_h^{h'}(s,a,s') - \iota \geqslant 64H^2\iota$ with probability $1 - \delta$.

It then holds that $P^{h'} \in \mathcal{P}^{h'}$ for each $h'$. Moreover, noting that for any $p \in \mathcal{P}^{h'}$ and $(h, s, a, s') \in \mathcal{W}^{h'}$, with similar computation it holds that

$$\left| p_{h,s,a,s'} - P_{h,s,a,s'}^{h'} \right| \leqslant \left| p_{h,s,a,s'} - \hat{p}_{h,s,a,s'}^{h'} \right| + \left| \hat{p}_{h,s,a,s'} - P_{h,s,a,s'}^{h'} \right|$$

$$\leqslant \frac{1}{3H}P_{h,s,a,s'}^{h'} + \frac{1}{3H}\hat{p}_{h,s,a,s'}^{h'}$$

$$\leqslant \left( \frac{2}{3H} + \frac{1}{9H^2} \right) P_{h,s,a,s'}^{h'}$$

As a result, $\mathcal{P}^{h'}$ is *tight* with respect to $P^{h'}$.

Fix $h \in [H]$. Recall that

$$\pi^{h,s,a} = \texttt{Policy Search}(\mathbf{1}_{h,s,a}, \mathcal{P}^{h-1});$$

$$\{\tilde{\pi}^h, p^h\} = \texttt{Sum}\left( \left\{ \frac{1}{SA}, \pi_{h,s,a}, p^h \right\}_{h,s,a} \right). \tag{51}$$

Recall that, for the first $h-1$ steps $\pi^h$ is the policy which is the same as $\tilde{\pi}^h$, and for the left $H - h + 1$ steps, $\pi^h$ is the uniformly random policy.

We first show that the $h$-th layer is well explored. By the property of $\texttt{Policy Search}$ and $\texttt{Sum}$ (see Lemma 8 and 2), there exists a constant $c > 0$ such that[12]

$$W^{\pi^{h,s,a}}(\mathbf{1}_{h,s,a}, P^{h-1}) \geqslant c \max_{\pi \in \Pi(r, \mathcal{P}^{h-1})} W^{\pi}(\mathbf{1}_{h,s,a}, P^{h-1})$$

$$W^{\pi^h}(\mathbf{1}_{h,s,a}, p^h) = \frac{1}{SA}W^{\pi_{h,s,a}}(\mathbf{1}_{h,s,a}, p^h), \forall (s,a) \in \mathcal{S} \times \mathcal{A}.$$

Noting that $p^h \in \mathcal{P}^{h-1}$ and $\mathcal{P}^{h-1}$ is *tight* with respect to $P^{h-1}$, by Lemma 17 we obtain that

$$W^{\pi^h}(\mathbf{1}_{h,s,a}, P^{h-1}) \geqslant \frac{c}{9SA} \max_{\pi \in \Pi(r, \mathcal{P}^{h-1})} W^{\pi}(\mathbf{1}_{h,s,a}, P^{h-1}) \tag{52}$$

for any $(s, a) \in \mathcal{S} \times \mathcal{A}$.

Using Lemma 4, with probability $1 - 4SA\delta$, the count of $(h, s, a)$ in the $h$-th batch is at least $\frac{ck}{27SA} \cdot \max_{\pi \in \Pi(r, \mathcal{P}^{h-1})} W^{\pi}(\mathbf{1}_{h,s,a}, P^{h-1}) - \iota$. As a result, we have that

$$N_h^h(s, a) \geqslant \frac{ck}{27SA} \cdot \max_{\pi \in \Pi(r, \mathcal{P}^{h-1})} W^{\pi}(\mathbf{1}_{h,s,a}, P^{h-1}) - 4\iota \tag{53}$$

for any $(s, a) \in \mathcal{S} \times \mathcal{A}$.

In the meantime, if $(h, s, a, s') \notin \mathcal{W}^h$, we have that $N_h^h(s, a, s') \leqslant C_1H^2\iota$. Using Lemma 4, with probability $1 - \delta$, we have that

$$kW^{\pi^h}(\mathbf{1}_{h,s,a}, P^{h-1})P_{h,s,a,s'} \leqslant 3C_1H^2\iota + 4\iota. \tag{54}$$

---

[12]We omit $\epsilon$ for convenience. By setting $\epsilon = 1/(SAHK)^{10}$, it is easy to verify the error only leads to a lower order term.

Combining (52) and (54), we have that

$$\max_{\pi \in \Pi(r, \mathcal{P}^{h-1})} \mathbb{P}_{\pi, P^{h-1}} \left[ (s_h, a_h, s_{h+1}) = (s, a, s') \right] = \max_{\pi \in \Pi(r, \mathcal{P}^{h-1})} W^{\pi}(\mathbf{1}_{h,s,a}, P^{h-1}) P_{h,s,a,s'}$$

$$\leqslant \frac{9SA}{c} W^{\pi^h}(\mathbf{1}_{h'',s,a}, P^{h-1}) P_{h'',s,a,s'}$$

$$\leqslant \frac{9SA}{c} \cdot \frac{3C_1 H^2 \iota + 4\iota}{k}$$

$$\leqslant \frac{36 C_1 SA H^2 \iota}{k}. \tag{55}$$

With an union bound over all $(h, s, a, s') \notin \mathcal{W}^h$, we have that

$$\max_{\pi \in \Pi(r, \mathcal{P}^{h-1})} \mathbb{P}_{\pi, P^{h-1}} \left[ (h, s_h, a_h, s_{h+1}) \notin \mathcal{W}^h \right] \leqslant \frac{36 C_1 S^2 A^2 H^2 \iota}{k}. \tag{56}$$

Note that $\mathcal{W}^h$ is non-decreasing in $h$. For any $\pi \in \cap_{h=1}^{h'} \Pi(r, \mathcal{P}^h)$, it holds that

$$\mathbb{P}_{\pi, P} \left[ \exists h' \leqslant H, (h', s_{h'}, a_{h'}, s_{h'+1}) \notin \mathcal{W}^H \right]$$

$$= \mathbb{P}_{\pi, P^H} \left[ \exists h' \leqslant H, (h', s_{h'}, a_{h'}, s_{h'+1}) \notin \mathcal{W}^H \right]$$

$$= \sum_{h'=1}^{H} \mathbb{P}_{\pi, P^H} \left[ (h', s_{h'}, a_{h'}, s_{h'+1}) \in \mathcal{W}^H, \forall 1 \leqslant h' < h, (h, s_h, a_h, s_{h+1}) \notin \mathcal{W}^H \right]$$

$$\leqslant \sum_{h=1}^{H} \max_{\pi \in \Pi(r, \mathcal{P}^{h-1})} \mathbb{P}_{\pi, P^{h-1}} \left[ (h, s_h, a_h, s_{h+1}) \notin \mathcal{W}^h \right]$$

$$\leqslant \frac{36 C_1 S^2 A^2 H^3 \iota}{k}. \tag{57}$$

Recall that $\Pi(r, \mathcal{P}) := \{\pi | U^{\pi}(r + \mathbf{1}_z, \mathcal{P}) \geqslant \max_{\pi} L^{\pi}(r, \mathcal{P})\}$. Because $P^h \in \mathcal{P}^h$ for any $h$, for any optimal policy $\pi^*$ and any policy $\pi'$, we have that $U^{\pi^*}(r + \mathbf{1}_z, \mathcal{P}) \geqslant V_1^*(s_1) \geqslant W^{\pi'}(r, P^h) \geqslant L^{\pi'}(r, \mathcal{P})$. Therefore, $\pi^* \in \Pi(\mathcal{P}^h)$ for any $1 \leqslant h \leqslant H$. By (57), (49) is proven.

In the case $u = 0$, we have that $\Pi(u, \mathcal{P}^h) = \overline{\Pi}$ for $1 \leqslant h \leqslant H$, where $\overline{\Pi}$ is the set of all possible policies. By (53), we have that

$$N_h^h(s, a) \tag{58}$$

$$\geqslant \frac{ck}{27SA} \max_{\pi} W^{\pi}(\mathbf{1}_{h,s,a}, P^h) - 4\iota \tag{59}$$

$$\geqslant \frac{ck}{27SA} \max_{\pi} W^{\pi}(\mathbf{1}_{h,s,a}, P) - 4\iota - \frac{ck}{27SA} \max_{\pi} \mathbb{P}_{\pi, P} \left[ \exists h' \in [H], (h', s_{h'}, a_{h'}, s_{h'+1}) \notin \mathcal{W}^H \right]$$

$$\geqslant \frac{ck}{27SA} \max_{\pi} W^{\pi}(\mathbf{1}_{h,s,a}, P) - 4\iota - \frac{36 C_1 SA H^3 \iota}{27}.$$

The proof is completed by noting that $N_h'(s, a) \geqslant N_h^h(s, a)$.

$\square$

### D.5  Statement and Proof of Lemma 17

**Lemma 17.** *Suppose $\mathcal{P}$ is* tight *with respect to $p$. Then we have that*

$$3W^{\pi}(\mathbf{1}_{h,s,a}, p) \geqslant W^{\pi}(\mathbf{1}_{h,s,a}, p') \geqslant \frac{1}{3} W^{\pi}(\mathbf{1}_{h,s,a}, p) \tag{60}$$

*for any $p' \in \mathcal{P}$, policy $\pi$ and $(h, s, a)$.*

*Proof.* For each trajectory $L = (s_1, a_1, ..., s_H, a_H, s_{H+1})$ such that $s_h \neq z$ for $1 \leqslant h \leqslant H + 1$, we have that

$$\mathbb{P}_{\pi, p}[L] = \pi_{h=1}^{H} \pi_h(a_h | s_h) p_{h, s_h, a_h, s_{h+1}} \geqslant e^{-\frac{H}{H}} \mathbb{P}_{\pi, p}[L] = \pi_{h=1}^{H} \pi_h(a_h | s_h) p'_{h, s_h, a_h, s_{h+1}} \geqslant \frac{1}{3} \mathbb{P}_{\pi, p'}[L].$$

So the left side of (60) is proven. By reversing $p$ and $p'$ the right side follows.

$\square$

# E    Other Missing Proofs

## E.1    Proof of Lemma 1

**Lemma 1 (restated)** *Let $d > 0$ be an integer. Let $\mathcal{X} \subset (\Delta^d)^m$. Then there exists a distribution $\mathcal{D}$ over $\mathcal{X}$, such that*

$$\max_{x = \{x_i\}_{i=1}^{dm} \in \mathcal{X}} \sum_{i=1}^{dm} \frac{x_i}{y_i} = md,$$

*where $y = \{y_i\}_{i=1}^{dm} = \mathbb{E}_{x \sim \mathcal{D}}[x]$. Moreover, if $\mathcal{X}$ has a boundary set $\partial \mathcal{X}$ with finite cardinality, we can find $\mathcal{D}$ in $\mathrm{poly}(|\partial \mathcal{X}|)$ time.*

*Proof.* Note that $\mathcal{X}$ is always bounded. Without loss of generality, we assume $\mathcal{X}$ is a discrete set with $\mathcal{X} = \{x^1, x^2, ..., x^L\}$ where $x^i = \{x_n^i\}_{n=1}^{dm}$ For $\lambda = \{\lambda_1, \lambda_2, ..., \lambda_L\} \in \Delta^L$, we define $E(\lambda)$ by

$$E(\lambda) := \Pi_{i=1}^{dm} \left( \sum_{j=1}^{L} \lambda_j x_i^j \right).$$

Then $E(\lambda)$ is bounded and $\Delta^L$ is compact. Consider to maximize $\ln(E(\lambda))$ over $\lambda \in \Delta^L$. It's not hard to verify that $\ln(E(\lambda))$ is concave in $\lambda$, so it is efficient to maximize it by gradient ascent algorithms. Let $\lambda^*$ be the optimal solution. By the KKT condition, we have that for any $j', j''$ such that $\lambda_{j'}^*, \lambda_{j''}^* \in (0, 1)$, it holds that

$$w := \sum_{i=1}^{dm} \frac{x_i^{j'}}{\sum_{j=1}^{} \lambda_j x_i^j} = \sum_{i=1}^{dm} \frac{x_i^{j''}}{\sum_{j=1}^{} \lambda_j x_i^j}.$$

Therefore, if for any $\lambda_j^* \neq 1$ for any $j$, we have that

$$w = \sum_{j=1}^{L} \lambda_j w = \sum_{i=1}^{dm} \frac{\sum_{j=1}^{} \lambda_j x_i^j}{\sum_{j=1}^{} \lambda_j x_i^j} = dm.$$

Then $\lambda^*$ is the desired solution. Otherwise, suppose $\lambda_1^* = 1$. Then we have that

$$dm = \sum_{i=1}^{dm} \frac{x_i^1}{x_i^1} \geqslant \sum_{i=1}^{dm} \frac{x_i^{j'}}{x_i^1}$$

for any $j' \geqslant 2$. Then $\lambda^*$ is also the desired solution. The proof is completed.    $\square$

## E.2    Proof of Lemma 2

**Lemma 2** (Restatement) *Let $\mathcal{P} = \otimes_{(h,s,a)} \mathcal{P}_{h,s,a}$ be a set of transition models such that $\mathcal{P}_{h,s,a} \subset \Delta^S$ is convex for any $(h, s, a)$. Let $\{(\pi^i, P^i)\}_{i=1}^{n}$ be a sequence of policy-transition pairs such that $P^i \in \mathcal{C}$. For any $\{\lambda_i\}_{i=1}^{n}$ such that $\lambda_i \geqslant 0$ for $i \geqslant 1$ and $\sum_i \lambda_i = 1$, there exists a policy $\pi$ and $P \in \mathcal{P}$, satisfying that*

$$W^\pi(\mathbf{1}_{h,s,a}, P) = \sum_i \lambda_i W^{\pi^i}(\mathbf{1}_{h,s,a}, P^i) \tag{61}$$

*for any $(h, s, a) \in [H] \times \mathcal{S} \times \mathcal{A}$. Furthermore, the time complexity to find $\{\pi, P\}$ could be bounded by $O(nS^3 A^2 H^2)$.*

*Proof.* By induction on $n$, it suffices to prove for the case $n = 2$. Our target is to find $(\pi, p)$ such that

$$W^\pi(\mathbf{1}_{h,s,a}, P) = \lambda_1 W^{\pi^1}(\mathbf{1}_{h,s,a}, P^1) + (1 - \lambda_1) W^{\pi^2}(\mathbf{1}_{h,s,a}, P^2) \tag{62}$$

holds for any $(h, s, a) \in [H] \times \mathcal{S} \times \mathcal{A}$. We will prove this by induction on $h$. For the case $h = 1$, since the initial distribution is fixed, we finish by letting

$$\pi_1(a|s) = \lambda_1 \pi_1^1(a|s) + (1 - \lambda_1)\pi_1^2(a|s) \tag{63}$$

for all $(s, a) \in \mathcal{S} \times \mathcal{A}$

Suppose (62) holds for any $1 \leqslant h' \leqslant h$ and any $(s, a) \in \mathcal{S} \times \mathcal{A}$. Then $\lambda_{h,s,a} = \frac{\lambda_1 W^{\pi^1}(\mathbf{1}_{h,s,a}, P^1)}{W^\pi(\mathbf{1}_{h,s,a}, P)}$ is well-defined. We set

$$P_{h,s,a} = \lambda_{h,s,a} P^1_{h,s,a} + (1 - \lambda_{h,s,a}) P^2_{h,s,a}$$

for any $(s, a)] \in \mathcal{S} \times \mathcal{A}$. By the inductive assumption

$$W^\pi(\mathbf{1}_{h,s,a}, P) = \lambda_1 W^{\pi^1}(\mathbf{1}_{h,s,a}, P^1) + (1 - \lambda_1) W^{\pi^2}(\mathbf{1}_{h,s,a}, P^2), \tag{64}$$

we have that for any $(s, a)$

$$P_{h,s,a} W^\pi(\mathbf{1}_{h,s,a}, P) = \lambda_1 W^{\pi^1}(\mathbf{1}_{h,s,a}, P^1) P^1_{h,s,a} + (1 - \lambda_1) W^{\pi^2}(\mathbf{1}_{h,s,a}, P^2) P^2_{h,s,a}.$$

We then have that

$$\lambda_1 \sum_{s',a'} W^{\pi^1}(\mathbf{1}_{h,s',a'}, P^1) P^1_{h,s',a',s} + (1 - \lambda_1) \sum_{s',a'} W^{\pi^2}(\mathbf{1}_{h,s',a'}, P^2) P^2_{h,s,',a',s}$$

$$= \sum_{s',a'} \lambda_{h,s',a'} W^\pi(\mathbf{1}_{h,s',a'}, P) P^1_{h,s',a',s} + \sum_{s',a'} (1 - \lambda_{h,s',a'}) W^\pi(\mathbf{1}_{h,s',a'}, P) P^2_{h,s',a',s} \tag{65}$$

$$= \sum_{s',a'} W^\pi(\mathbf{1}_{h,s',a'}, P) P_{h,s',a',s}, \tag{66}$$

which implies that

$$W^\pi(\mathbf{1}_{h+1,s}, P) = \lambda_1 W^{\pi^1}(\mathbf{1}_{h+1,s} P^1) + (1 - \lambda_1) W^{\pi^2}(\mathbf{1}_{h+1,s} P^2) \tag{67}$$

for any $s \in \mathcal{S}$, where the reward function $\mathbf{1}_{h+1,s} = \sum_a \mathbf{1}_{h+1,s,a}$. Let

$$\pi_{h+1}(a|s) = \frac{\lambda_1 W^{\pi^1}(\mathbf{1}_{h+1,s}, P^1) \pi_{h+1}^1(a|s) + (1 - \lambda_1) W^{\pi^2}(\mathbf{1}_{h+1,s}, P^2) \pi_{h+1}^2(a|s)}{W^\pi(\mathbf{1}_{h+1,s}, P)}. \tag{68}$$

Then it is easy to verify that

$$W^\pi(\mathbf{1}_{h+1,s,a}, P) = W^\pi(\mathbf{1}_{h+1,s}, P) \pi_{h+1}(a|s) = \lambda_1 W^{\pi^1}(\mathbf{1}_{h+1,s,a}, P^1) + (1 - \lambda_1) W^{\pi^2}(\mathbf{1}_{h+1,s,a}, P^2).$$

Also note that the process above costs at most $O(S^3 A^2 H^2)$ time, so the total computational cost is bounded by $O(nS^3 A^2 H^2)$. The proof is completed. $\qquad\square$