# OpenReview forum: "Near-Optimal Regret Bounds for Multi-batch Reinforcement Learning"
_NeurIPS.cc/2022/Conference — NeurIPS 2022 Accept_

### Official Review · Reviewer_jpiL · 2022-06-27

**Rating:** 6
**Confidence:** 4
**Soundness:** 3 good
**Presentation:** 2 fair
**Contribution:** 3 good

**Summary:**

The paper studies a batched reinforcement learning (RL) problem where the interaction is divided into $M$ epochs (‘batches’). The agent chooses the batch lengths before the interaction starts and can only update its policy at the beginning of a new batch. This setting is well-motivated in real-world applications where policies are deployed and adaptive/frequent policy updates are undesirable. As in the multi-armed bandit (MAB) problem, the authors show that any minimax algorithm must use $M=\Omega(\log\log K)$ policy batches (where $K$ is the total number of episodes). They complement these lower bounds by proving a tight regret upper bound for a policy elimination algorithm that uses $M=H+\log\log K$ batches.

**Questions:**

Questions:
* Why do you need two exploration stages?
* Why do you need to use the confidence bounds for each next state separately, instead of the standard L1 bounds for the transition kernel? Is it due to the tightness requirement from the transition region? If so, can’t it be modified to allow these confidence intervals? It feels as if this is the main reason for the raw exploration stages.

_____

Minor/other comments (notice the numerous typos):
* Definition 1 – the summation should end at $M$.
* Theorem 1 – please move the footnote location – it looks like a power.
* Line 116 – did you mean [N]?
* Line 119  -a missing space before the citations.
* Line 135 – variance of what?
* Line 139 – after a batch, shouldn’t it be $N_h(s,a)\approx d^{\pi}(s,a) t^m$?
* Line 165 – “by by”
* The derivation in lines 176-177 wasn’t very clear.
* I would also recommend putting more abstract algorithm blocks in the main paper, and leaving the detailed algorithm to the appendix.
* Line 200 – delete “...such that $\tilde{\pi}\in\Pi$”.

### Post-rebuttal
I thank the authors for the response.

I am sorry for the confusion - I of course meant that the initial stage requires knowing the number of episodes $K$, and not $H$ (which is a completely different setting). As I said, it is a minor issue.

**Limitations:**

I would have liked to see a more concrete discussion on the computational complexity of the algorithm – and not just O-notations

**Strengths And Weaknesses:**

I think that the framework is interesting and the results are nontrivial. Also, to the best of my knowledge, the algorithmic design is novel – while successive elimination algorithms are common in MABs, I am not familiar with (polynomial) SE algorithms in RL that eliminate in the transition space.

On the con side, the resulting algorithm is *very* complicated, uses numerous subroutines, each with its own computational complexity, and it is quite hard to understand its overall computational complexity (even if it’s polynomial). In general, I think that the paper was hard to follow, and its clarity could be improved. Another (minor) weakness is that the initial exploration stage requires knowing the horizon – it’s unclear whether the algorithm can be converted to an anytime algorithm.

---

> ### Author Response · Authors · 2022-08-02
> **Response to Reviewer jpiL**
>
> We thank the reviewer for valuable feedback. We appreciate the suggestions on writings and will improve accordingly. Below we present our response:
>
> > Another (minor) weakness is that the initial exploration stage requires knowing the horizon – it’s unclear whether the algorithm can be converted to an anytime algorithm.
>
> In this case the planning horizon $H$ is unknown, the learner may pay one episode to learn the value of $H$. If $H$ is a random variable independent of the policy $\pi$, the learner can also learn the value of $H$ in $O(\sqrt{K})$ episodes. There are also some cases with some ending states (e.g., the states in the $H$-th layer are ending states in the original problem) . Therefore the planning horizon may depend on the policy $\pi$. In this case, the worst case regret depends on the maximal possible planning horizon $H_{max}$, and we think a doubling method may help to solve the problem.
>
> > Why do you need two exploration stages?
>
> The main reason is to bound the leading term in the regret by $\tilde{O}(\sqrt{SAKH^3\iota})$. If we only use one exploration phase, the regret bound would be $\max\\{Hk_1,  \frac{TS^3A^3H^4\iota}{k_1}\\}$, which is larger than $\tilde{O}(\sqrt{SAKH^3\iota})$.
>
> > Why do you need to use the confidence bounds for each next state separately, instead of the standard $L_1$ bounds for the transition kernel? Is it due to the tightness requirement from the transition region? If so, can’t it be modified to allow these confidence intervals? It feels as if this is the main reason for the raw exploration stages.
>
> It is necessary to bound each $P_{h,s,a,s’}$ separately. In the case $P_{h,s,a,s’}$ is very close to $0$, $L_1$ bounds do not provide useful information about $P_{h,s,,a,s’}$. On the other side, separate bounds can learn a \emph{tight} estimation of $P_{h,s,a,s’}$ efficiently.
>
> > I would have liked to see a more concrete discussion on the computational complexity of the algorithm – and not just O-notations
>
> Please refer to the response about computational cost.
>
>
>
> > Line 116 – did you mean $[N]$?
>
> Yes.
>
> > Line 135 – variance of what?
>
> The variance of $P_{h,s,a}$ with respect to $V^*_{h+1}$, i.e, the variance of X such that $P[X=V^*_{h+1}(s’)]=P_{h,s,a,s’}$;
>
> >Line 139 – after a batch, shouldn’t it be $N_h(s,a)\approx d^{\pi}(s,a)t^m$?
>
> Thanks to point this out. We want to explain that $N_h(s,a) \propto d^{\pi}_h(s,a)$.
>
> >The derivation in lines 176-177 wasn’t very clear.
>
> We will add a line to explain this inequality.
> Here we use the definition of $r^{(i)}$ and the fact that: for  $x_i\in [0,1]$, $i=1,2,...,n$,  $\sum_{i=1}^n x_i \cdot \min \\{        \frac{1}{\sum_{j=1}^{i-1}x_j}, 1\\}\leq \sum_{i=1}^n \frac{2x_i}{ \sum_{j=1}^{i-1}x_j+1} \leq 4\sum_{i=1}^n \ln( \frac{\sum_{j=1}^{i}x_j+1}{\sum_{j=1}^{i-1}x_j+1})\leq 4 \ln(\sum_{j=1}^{n}x_j+1)\leq 4\ln(n+1)$.

---

### Official Review · Reviewer_gMzA · 2022-06-27

**Rating:** 7
**Confidence:** 4
**Soundness:** 3 good
**Presentation:** 3 good
**Contribution:** 3 good

**Summary:**

This paper studies multi-batch reinforcement learning in the finite-horizon MDPs. They propose an algorithm that achieves near-optimal regret bound $O(\sqrt{SAH^3K})$ in $K$ episodes using only $O(H + \log_2\log_2K)$ batches. They also show a nearly matching lower bound stating that in order to achieve $O(\text{poly}(S, A, H)\sqrt{K})$ regret, the number of batches is at least $\Omega(H/\log_A(K) + \log_2\log_2K)$.

**Questions:**

1. Is the overall algorithm computationally efficient? The authors address this question for some component of their algorithm but does not mention the overall computation complexity. The name "policy elimination" seems to suggest that it is computationally inefficient, but the pseudo code does not seem to contain computationally inefficient components. A clarification on this would be useful.
2. Is there any potential way to simplify the algorithm?
3. Why is EVI not enough for policy search?

**Limitations:**

There are several typos:
1. Definition 1: $\sum_{m=1}^Kt_m$
2. Equation (2): missing $p$
3. line 163: double "by"

**Strengths And Weaknesses:**

Strength: the problem studied in this paper is very important and has significant practical values. The proposed algorithm has a strong theoretical performance that nearly matching the lower bound. There are also several non-trivial technical contributions in this paper.

Weakness: the final algorithm is fairly complicated and hard to follow. My suggestion is that the author should spend more paragraphs on discussing the algorithm design, and postpone the proof sketch to Appendix. Discussion on why existing approaches achieve sub-optimal results, what are the challenges, and how each component of this paper tackle these challenges would be very helpful for the readers to appreciate the contributions of this paper.

---

> ### Author Response · Authors · 2022-08-02
> **Response to Reviewer gMzA**
>
> Thanks for the suggestions. Below we present our responses.
>
> > The final algorithm is fairly complicated and hard to follow. My suggestion is that the author should spend more paragraphs on discussing the algorithm design, and postpone the proof sketch to Appendix.
>
>  We will add more explanation about our algorithm.
>
> At a higher level, our algorithm consists of an exploration phase and a policy-elimination phase. The target of the exploration phase is to learn an approximate transition model, which is set as the input to the policy-elimination phase. And in the policy-elimination phase, we compute the optimal-design policy with the help of this approximate transition model. Moreover, to reduce the regret bound and computational cost, we make some improvements over the framework mentioned above.  For example, we compute the value function $\{v_h^{m}(s)\}$ to help reduce the variance.
>
> > Discussion on why existing approaches achieve sub-optimal results, what are the challenges, and how each component of this paper tackle these challenges would be very helpful for the readers to appreciate the contributions of this paper.
>
>  Previous methods mainly follow the doubling trick to update the policy, which leads to $O(SAH\log_2(K))$ switches of policies. Such methods are always optimistic learning methods, which is similar to UCB in bandit algorithms. As a result, the batch complexity is far from tight.
>
> In this work, our main technique contribution is proposing the near-optimal design scheme, which helps us to generalize the policy-elimination framework from bandits to RL.
> The left efforts are devoted to computationally efficient implementation, where we have several points: (1) reward-zero learning to compute the approximate optimal design; (2) approximate transition model to help estimate the occupancy distribution $d^{\pi}_h(\cdot,\cdot)$; (3) efficient implementation of constrained policy search.
>
> > Is the overall algorithm computationally efficient? The authors address this question for some component of their algorithm but does not mention the overall computation complexity. The name "policy elimination" seems to suggest that it is computationally inefficient, but the pseudo code does not seem to contain computationally inefficient components. A clarification on this would be useful.
>
> Please refer to the response about computational cost.
>
> > Is there any potential way to simplify the algorithm?
>
> We find it is hard to simplify the algorithm while keeping the computational cost affordable. We will consider presenting a warm-up algorithm with exponential computational cost. We believe such an algorithm would be simpler and could provide the insights of the final algorithm.
>
> > Why is EVI not enough for policy search?
>
> EVI can learn the optimal value function and corresponding optimal policy without constraints. However, for constrained policy search, EVI does not work simply.

---

### Official Review · Reviewer_pfXX · 2022-07-11

**Rating:** 6
**Confidence:** 4
**Soundness:** 2 fair
**Presentation:** 1 poor
**Contribution:** 3 good

**Summary:**

In this paper, this paper studies the episodic reinforcement learning (RL) problem modeled by finite-horizon Markov Decision Processes (MDPs) with constraints on the number of batches. It achieves near-optimal regret $O(\sqrt{SAH^3K})$ using $O(H+\log\log(K))$ batches. The technical contribution includes a near-optimal design scheme to explore the unlearned states and a computationally efficient algorithm to explore certain directions with an approximated transition model.

**Questions:**

The writing of the paper requires significant improvement. Concretely, see my comments in above.

**Limitations:**

Yes.

**Strengths And Weaknesses:**

**Strengths**: On the technique ends, I think this paper provides some solid contribution to achieving near-optimal regret, In particular, Lemma 1 leverages a G-optimal design style result for offline reinforcement learning and the proof is delicate. In addition, the decomposition (34) is critical for achieving tight regret with the term T2 can be bounded as a higher-order term. However, there are a few aspects that affect the quality of this paper.

**Weaknesses**:

1. The writing is not professional. There are numerous parts that are unclear. See the following.

2. the value of $k_1=\sqrt{SAKH}$ would deteriorate the the claimed dependence of $\sqrt{H^3}$ in regret. Raw exploration will become the dominant term and would make the bound has order $\sqrt{H^5}$.

3. Lemma 4 seems to be inconsistent with Lemma 4 (restated).

4. The equation in line 582 is not accurate since $P^{h'}$ is population-level quantities.

5. equation (52) seems to be wreak havoc, also equation (54).

6. Also, line 617-622 seems disconnected.

---

> ### Author Response · Authors · 2022-08-02
> **Response to Reviewer pfXX**
>
> We thank the reviewer for valuable feedback. We apologize to the typos and mistakes. Below we fix the mistakes accordingly.
>
> > The value of $k_1 = \sqrt{SAKH}$ would deteriorate the the claimed dependence of  $\sqrt{H^3}$ in regret. Raw exploration will become the dominant term and would make the bound has order $\sqrt{H^5}$.
>
>  We should choose $k_1 = 144\sqrt{SAK\iota/H}$. The lower order terms in the final regret bound are also updated in the new version.
>
> > Lemma 4 seems to be inconsistent with Lemma 4 (restated).
>
>    We rewrite Lemma 4 as the restated version;
>
> > The equation in line 582 is not accurate since  $P^{h'}$ is population-level quantities.
>
>  We add a line in the new version to show that $N_h^{h’}(s,a)P_{h,s,a,s’}^{h’} \geq \frac{1}{3}N^{h’}_h(s,a,s’) -\iota \geq 64H^2\iota$.
>
> Then we can bound the error between empirical transition model and the clipped true transition model by $\frac{1}{3H}P_{h,s,a,s’}^{h’}$;
>
>
> > Equation (52) seems to be wreak havoc, also Equation (54).
>
>  We fix the equations.
>
> > Also, line 617-622 seems disconnected.
>    This part is redundant and is deleted in the latest version.

---

> > ### Comment · Reviewer_pfXX · 2022-08-09
> > **Further reply**
> >
> > Thank you for the response. I am happy to keep my original score. In addition, I believe **Sample-Efficient Reinforcement Learning with loglog(T) Switching Cost** seems to be a paper that is very related to this paper. The authors should include discussion about this paper.

---

> > > ### Author Response · Authors · 2022-08-09
> > > **Response to Reviewer pfXX**
> > >
> > > Thank you for the suggestion.  [Qiao et. al.] works on a similar task, but they mainly consider deterministic policies, while the policies in our algorithm might be non-deterministic. We will carefully discuss this paper in the related works.

---

### Official Review · Reviewer_YiLt · 2022-07-11

**Rating:** 5
**Confidence:** 3
**Soundness:** 3 good
**Presentation:** 2 fair
**Contribution:** 3 good

**Summary:**

Authors propose a novel algorithm for multi-batch reinforcement learning with nearly minimax optimal regret and $O(H + \log\log K)$ batch complexity. Additionally, authors propose polynomial time implementation, and provide a lower bound on batch complexity of order $\Theta(H/\log_A(K) + \log \log K)$ to show that their algorithm is nearly optimal in this parameter too.

**Questions:**

1) In the main paper there is a lack of intuition how algorithm achieves such a small batch complexity as $O(H + \log \log K)$, whereas it seems to be one of the most crucial points of the proposed algorithm. It would be great to see such an intuition.

2) Additionally, it would be very interesting to see the full computational complexity of the algorithm and a more direct description on the places inside the algorithm where new batch starts;

Additionally, there are several typos was notices and needs to be fixed

- Line 78: Typo in asymptotic;
- Line 107: Wrong indexes in the definition of V- and Q-functions;
- Line 123: missing p in the definition of lower bound;
- Line 163: double “by”
- Line 492: formatting problems with the table of notation; lack of important notation in this table such as $W(v, p)$.
- Line 598: strange line break that makes formula unreadable;
- Line 617,618: broken links to Proposition and Lemma.

**Limitations:**

Authors addressed all the limitations of their work.

**Strengths And Weaknesses:**

Strengths:

1) The setting of multi-batch RL is seems to be very valuable in the modern RL community and up to my knowledge it is the first theoretical result in this sphere;

2) Nearly optimal regret bound in the first term and nearly optimal batch complexity; iterated logarithm of policy switches sounds to be very interesting even in a simpler setting of counting the number of policy switches.

3) Provided lower bound to strengthen the result on batch complexity;

Weaknesses:

1) Whereas the setting of multi-batch RL is interesting in applications, the proposed algorithm is seems to be impossible to implement beyond tabular setting due to policy elimination structure;

2) In the main paper there is no mentioned exact computational complexity of the algorithm, only the fact that it is polynomial; In the algorithm description we may see that the algorithm needs to take a loop over $K^3$ elements, that make the computation complexity enormous at least on one stage of the algorithm.

3) Not very good dependence on a number of states in second-order terms; It seems that the regime of nearly minimax optimality of this algorithm appears only after $K \geq S^{12} A^{8} H^{24}$ (to make the first-order regret term greater than the third one).

4) Hard-to-follow description of the algorithm both as a pseudo-code and as a text description.

---

> ### Author Response · Authors · 2022-08-02
> **Response to Reviewer YiLt**
>
> Thanks for your valuable comments. Below we present our response to the comments.
>
> >Whereas the setting of multi-batch RL is interesting in applications, the proposed algorithm is seems to be impossible to implement beyond tabular setting due to policy elimination structure
>
> Our algorithm is based on policy elimination in high-level intuition, while we implement policy elimination implicitly by learning a constrained RL problem, i.e., find a survived policy $\pi$ such that $W^{\pi}(u,P)\geq c W^{\pi’}(u,P)-\epsilon$ for any survived policy $\pi’$. We believe such methods could be generalized to more complicated settings like linear MDP.
>
> > In the main paper there is no mentioned exact computational complexity of the algorithm, only the fact that it is polynomial; In the algorithm description we may see that the algorithm needs to take a loop over  $K^3$  elements, that make the computation complexity enormous at least on one stage of the algorithm.
>
> Please refer to the response about the computational cost.
>
> > Not very good dependence on a number of states in second-order terms; It seems that the regime of nearly minimax optimality of this algorithm appears only after $K\geq S^{12}A^8H^{24}$  (to make the first-order regret term greater than the third one).
>
> We agree that the second order term may not be optimal, which is mainly due to the complex techniques employed to ensure the near-optimal leading term. In our algorithm, to keep the leading term near-optimal, we use two exploration phases to deal with the infrequent terms, which leads to a large amount of small order terms. Besides, we also use the value function in the previous batch to reduce the variance, which leads to some more complicated lower order terms.
>
> The main focus of this paper is to present the first asymptotically near-optimal RL algorithm in the batch learning setting, and we leave the optimization of the lower order terms as future research directions. Indeed, even for the ordinary tabular RL problem (without the batch constraints), it takes quite a few follow-up works to optimize the lower order regret terms (e.g., [Li et al., 2021, Menard et al., 2021, Zhang et al., 2021]).
>
> We will add a paragraph in the conclusion section to discuss this future direction and possible solutions. For example, a simpler analysis would show that with the batch constraints, the regret can be at most $\tilde{O}(k_2H^2+\sqrt{S^2AH^3K})$, which is better in the regime of smaller $K$.
>
> References:
>
> Li, Gen, et al. "Sample complexity of asynchronous Q-learning: Sharper analysis and variance reduction." Advances in neural information processing systems 33 (2020): 7031-7043.
>
> Ménard, Pierre, et al. "UCB Momentum Q-learning: Correcting the bias without forgetting." International Conference on Machine Learning. PMLR, 2021.
>
> Zhang, Zihan, Xiangyang Ji, and Simon Du. "Is reinforcement learning more difficult than bandits? a near-optimal algorithm escaping the curse of horizon." Conference on Learning Theory. PMLR, 2021.
>
>
> > Hard-to-follow description of the algorithm both as a pseudo-code and as a text description
>
> We have fixed the typos in the new version. We will add a paragraph to explain the  high-level intuitions in Section 5.
>
> > Question 1:
> In the main paper there is a lack of intuition how algorithm achieves such a small batch complexity as , whereas it seems to be one of the most crucial points of the proposed algorithm. It would be great to see such an intuition.
>
> The proposed algorithm follows the idea of policy elimination in bandit problems. In bandit problems, the learner takes the survived arms with equal probability in one batch. Indeed we generalize this intuition to the setting of RL, which could be viewed as a linear optimization problem (i.e., to max $\sum_{s,a,h}d^{\pi}_h(s,a)r_h(s,a)$). By results for bandit problems, it is natural to obtain a batch complexity of $\log_2\log_2(K)$ for this part. On the other hand, to learn a good policy, it is natural to ask the learner to visit each state-action pair for at least one time (if possible). So $\Omega(H/log_A(K))$ batches are necessary to make this. The corresponding algorithm is also natural: one just needs to explore layer by layer inductively.

---

> > ### Comment · Reviewer_YiLt · 2022-08-08
> > **Thank you for response**
> >
> > Thank you for your reply!
> >
> > After reading all other reviews and responses, I still think that it is an interesting contribution to the RL community. I would like to keep my score.

---

### Author Response · Authors · 2022-08-02
**Response about the exact computational cost**

We first thank the reviewers for valuable feedback. A common concern of the reviewers is the exact computational cost. We answer this question as below.

Conclusion: the total computational cost is $\tilde{O}(S^4AHK^3 + S^3A^2H^2K^3)$.

Analysis: we first consider the computational cost of two important subroutines $\mathtt{Sum}$ and $\mathtt{PolicySearch}$.

 (a) Computational cost for $\mathtt{Sum}$: in the case the number of inputs $n=2$, following the arguments in the proof of Lemma 2, the computational cost is $O(S^3A^2H^2)$. For general $n$, we can implement $\mathtt{Sum}$ with $2$ inputs for $n$ times. Therefore, the computational cost is $O(nS^3A^2H^2)$.

(b) Computational cost of $\mathtt{PolicySearch}$: as described in Algorithm 4, to compute $a$ and $b$. we need to invoke EVI(extended value iteration) for two times. Besides, EVI is invoked for at most $\log(1/(a-b)\epsilon)=O(\log(SAHK))$ times to compute $(\pi^{(i)}, P^{(i)})$ and $\mathtt{Sum}$ is used for one time with 2 inputs. Now we analyze the computational cost of EVI. By definition, EVI consists of  $SAH$ constrained programming (i.e., $\max_{p\in \mathcal{P}}pV_{h+1}$ ) . Because $\mathcal{P}$ could be written as  $\mathcal{P} =  \\{   p\in \Delta^{S} | p^{\top} a_i \leq b_i   \\}$, we can view this problem as linear programming. There are at most $O(SM)$ constraints because in each batch the number of new constraints for $\mathcal{P}$ is at most $S+1$. By  the results in [Cohen, Lee and Song], to find an $\epsilon$-optimal solution, the computational cost is $O((SM)^{3}\log(SM/\epsilon)) $. Recall that $\epsilon = 1/(SAKH)^{10}$, the total computational cost of EVI is bounded by $O(S^4AHM^3\log(SAKH))$. Then the computational cost of $\mathtt{PolicySearch}$ is $O(S^4AHM^3\log^2(SAKH))$.

Now we count the number of callings to  $\mathtt{Sum}$ and $\mathtt{PolicySearch}$.

In the first and second stage, $\mathtt{Sum}$ is called for $2H$ times with $SAH$ inputs,  and $\mathtt{PolicySearch}$ is called for $2H$ times; in the third stage, $\mathtt{Sum}$ is called for $M$ times with $K^3$ components, and $\mathtt{PolicySearch}$ is called for $K^3M$ times. Besides, EVI is called for one time to compute the optimal value function $\\{ v^m_h(\cdot)\\}$ for each batch in the third stage. So the total computational cost due to $\mathtt{Sum}$, $\mathtt{PolicySearch}$ and EVI is bounded by $\tilde{O}(S^4AHK^3 + S^3A^2H^2K^3)$. Besides, to observe the samples and compute the confidence region, we need $O(S^2AHK)$ time. So the total computational cost is  $\tilde{O}(S^4AHK^3 + S^3A^2H^2K^3)$.

Moreover, we  can also choose $k =K$ in Equation (4) (this will lead to an additional term in the lower order terms). In this way, the total computational cost is $\tilde{O}(S^4AHK + S^3A^2H^2K)$.

Refernece: Cohen M B, Lee Y T, Song Z. Solving linear programs in the current matrix multiplication time[J]. Journal of the ACM (JACM), 2021, 68(1): 1-39.

---

### Meta-Review · Area_Chair_6r8W · 2022-08-27

**Recommendation:** Accept
**Confidence:** Certain

**Metareview:**

The paper studies the batch RL problem, in which the algorithm first decide a switching schedule and then switch the policies based on this schedule. The proposed approach achieves a good regret upper bound matching existing non-batch algorithms (although the lower-order terms are still large). The batch complexity on the other hand matches the lower bound (up to log factors).

The reviewers believe that the theoretical contributions are solid and qualified to be published in NeurIPS. The authors did a good a job in addressing the computation complexity in the rebuttal phase. The meta-reviewer suggests the authors to further clarify presentation issues. Also, it would be good to cite the recent RL theory papers in the tabular setting (including those with a generative model).

**Award:**

No

---

### Decision · Program_Chairs · 2022-09-14

Accept